# Budget-Efficient Attacks and Robustness Training for Cooperative MARL

**Junyong Jiang**[1]  **Xin Yuan**[2]  **Longhe Lin**[1]  **Songze Li**[1 3]  **Lu Dong**[1 3 *]

## Abstract

Cooperative multi-agent reinforcement learning (CMARL) policies are vulnerable to action hijacking even when only a few timesteps are compromised. Recent adversarial attacks and adversarial training methods have been explored, but under an explicit attack budget, existing attacks often fail to accurately expose critical coordination weaknesses and incur substantial training cost. We propose Budgeted Hierarchical Efficient Attack (BHEA), a budgeted hierarchical adversarial attack that separates decisions on when and which agents to hijack from action replacement, enabling more precise vulnerability discovery under limited attack opportunities. We further show that training cooperative policies against BHEA substantially improves robustness to limited-step action hijacking while reducing training overhead. Experiments on the Star-Craft Multi-Agent Challenge (SMAC) and Multi-Agent Particle Environments (MPE) demonstrate stronger attacks under the same attack budget and improved robustness. Code is available at https://github.com/ji6ng/BHEA.

## 1. Introduction

CMARL has demonstrated remarkable efficacy in orchestrating complex coordination tasks across diverse and challenging environments (Vinyals et al., 2019; Kuba et al., 2022; Wen et al., 2022; Zhou et al., 2025). While state-of-the-art algorithms have achieved superhuman performance in challenging benchmarks such as the SMAC (Sunehag et al., 2018; Foerster et al., 2018; Rashid et al., 2020; Wang et al., 2021), the resulting cooperative policies often exhibit se-

vere fragility in deployment. The malfunction or adversarial compromise of even a single agent can trigger cascading coordination failures, precipitating catastrophic performance degradation (Gleave et al., 2020; Liu et al., 2024; Liu & Lai, 2023). This inherent brittleness presents a fundamental barrier to deploying MARL systems in safety-critical or adversarial environments where reliability is paramount.

Although adversarial training has been widely adopted to improve robustness in CMARL (Standen et al., 2025; Zhou et al., 2024; Ma & Li, 2025), existing defenses often fail when agent actions are compromised at only a small number of critical timesteps. A key reason is that the adversarial attacks used during training are often inefficient, requiring repeated attacks or extensive search to achieve strong effects. In practice, commonly used attacks either rely on planner-based mechanisms that repeatedly target selected agents using explicit attack logic, as in Wolfpack (Lee et al., 2025), or adopt population-based evolutionary search to generate diverse adversarial policies, as in EGA (Yuan et al., 2023). Both approaches incur substantial training cost and achieve limited attack effectiveness when only a small number of attacks are applied. At the same time, standard flat adversarial policies struggle to jointly reason about *when* to attack, *whom* to target, and *how* to replace the hijacked actions, as these tightly coupled decisions must be optimized over a vast combinatorial space (Liu et al., 2024; Cheng et al., 2023; Guo et al., 2021b). As a result, the adversarial signals provided during training are often insufficiently strong or targeted, leading to only modest robustness improvements in the limited-step regime and motivating a rethinking of adversarial attack design with an emphasis on attack efficiency and adaptive decision-making.

In this paper, we address these challenges by proposing BHEA, a strong and budget-efficient adversarial attack for CMARL. BHEA is specifically designed to maximize disruption under explicit attack budgets, where only a limited number of timesteps and agents can be compromised. Rather than jointly optimizing attack timing, victim selection, and action manipulation within a single flat policy, BHEA adopts a hierarchical structure that separates high-level decisions on *when* and *which* agents to hijack from low-level action replacement. This decomposition enables explicit reasoning over scarce attack opportunities and coordinated victim selection, resulting in substantially stronger

[1]School of Cyber Science and Engineering, Southeast University, Nanjing, Jiangsu, China [2]School of Automation, Southeast University, Nanjing, Jiangsu, China [3]Engineering Research Center of Blockchain Application, Supervision and Management (Southeast University), Ministry of Education, China. Correspondence to: Lu Dong <ldong90@seu.edu.cn>.

*Proceedings of the 43rd International Conference on Machine Learning*, Seoul, South Korea. PMLR 306, 2026. Copyright 2026 by the author(s).

attacks that induce greater disruption with fewer attack steps than prior methods.

Building on this stronger and more budget-efficient attack, we further demonstrate its effectiveness as a training signal for robustness. Specifically, we develop BHEA-AT, an adversarial training framework that uses BHEA as a training opponent to expose critical coordination failures. Training against such an efficient attacker encourages cooperative policies to learn recovery strategies and coordination-preserving behaviors under short and sporadic compromises. Importantly, the resulting robustness improvements arise as a direct consequence of training against a stronger attack, while avoiding the computational overhead and complexity commonly associated with existing adversarial training frameworks.

We evaluate our approach on standard SMAC and MPE benchmarks. The results show that BHEA consistently induces stronger performance degradation than prior attacks under the same number of attack steps, and that training against BHEA substantially improves robustness to limited-step action hijacking while significantly reducing training cost. These findings highlight the importance of budget-aware adversarial attacks as an effective and practical foundation for robust CMARL.

In summary, this paper makes the following contributions:

- We propose BHEA, a strong and budget-efficient adversarial attack for CMARL that induces substantially greater disruption with fewer attack steps than existing methods.

- Meanwhile training against BHEA improves robustness to action hijacking, particularly under limited-step compromises where only a small number of critical timesteps are attacked.

- Our approach simplifies adversarial training and substantially reduces computational overhead, achieving strong robustness while requiring significantly less training time than prior adversarial training methods.

## 2. Related Work

**Adversarial Attacks in MARL.** Research on adversarial attacks in MARL is commonly organized by the attack surface being exploited. A large body of work studies input-side attacks, including perturbations to observations and communication channels, with the goal of misleading agents' perception and coordination (Zhang et al., 2020; Ilahi et al., 2021; Zhou et al., 2023; Tekgul et al., 2022; Qiaoben et al., 2024). Another line of research considers execution-time attacks that interfere with the actions executed by agents, rather than corrupting their inputs (Sun

et al., 2020; Mo et al., 2022; Bukharin et al., 2023; Liu & Lai, 2023). Orthogonal to this distinction, several works focus on learning adversarial agents or policies that interact with the victim team to induce failures through strategically chosen behaviors (Gleave et al., 2020; Guo et al., 2021a; Wu et al., 2021; Li et al., 2025).

Within execution-time attacks, action hijacking refers to directly overriding agent actions during execution. Existing approaches in this setting mainly adopt planning-based or population-based strategies. Planner-based methods such as Wolfpack (Lee et al., 2025) use explicit procedural planning to identify critical timesteps and victim groups, while population-based methods such as EGA (Yuan et al., 2023) rely on evolutionary search over a population of adversarial policies. Empirically, both approaches depend on substantial planning or population search to generate strong attacks, yet exhibit limited effectiveness when only a small number of hijacking steps are applied.

**Adversarial Training and Robust MARL.** Adversarial training is a standard defense mechanism in reinforcement learning, formulated as a minimax game where agents optimize against disturbances or opponents(Oikarinen et al., 2021; Zan et al., 2023; Alqahtani & Halabi, 2023; Liang et al., 2022). Foundational works in single-agent domains, such as RARL (Pinto et al., 2017), demonstrated that training against a learned adversary can significantly enhance robustness to modeling errors. Extending this to CMARL, recent approaches have focused on population-based adversarial training to mitigate the risk of overfitting to a single attacker. Prominent examples include RAP (Vinitsky et al., 2020), which maintains a diverse population of adversaries, and ROMANCE (Yuan et al., 2023), which employs evolutionary strategies to generate high-quality auxiliary attackers. More specific defenses like WALL (Lee et al., 2025) have been proposed to counter coordinated multi-agent attacks by fostering system-wide collaboration. However, the efficacy of these methods is fundamentally bounded by the optimality of the adversarial inner loop. When attacks are applied sparsely, existing planner-based or population-based attackers often rely on repeated attacks or extensive search, yielding weak training signals and limited robustness against precise action hijacking. Additional background on planner-based and population-based attackers (e.g., Wolfpack and EGA) as well as their corresponding robust training frameworks (WALL and ROMANCE), and their relation to our formulation, is provided in Appendix B.

## 3. Problem Formulation and Threat Model

### 3.1. CTDE-Based CMARL

We consider a CMARL problem under the Centralized Training with Decentralized Execution (CTDE) (Lowe et al.,

2017) paradigm. The environment is modeled as a decentralized partially observable Markov decision process (Dec-POMDP), defined by the tuple

$$\mathcal{M} = \langle N, \mathcal{S}, \{\mathcal{A}^i\}_{i=1}^N, P, \mathcal{O}, O, R, \gamma \rangle,$$

where $N$ denotes the number of agents, $\mathcal{S}$ the global state space, $\mathcal{A}^i$ the action space of agent $i$, $P(s_{t+1} \mid s_t, \mathbf{a}_t)$ the transition function, $\mathcal{O}$ the observation space, $O(o_t^i \mid s_t, i)$ the observation function, $R(s_t, \mathbf{a}_t)$ the shared team reward, and $\gamma \in [0, 1)$ the discount factor.

At each timestep $t$, agent $i$ receives a local observation $o_t^i$ and selects an action $a_t^i$ according to its decentralized policy $\pi_v^i(a_t^i \mid \tau_t^i)$, where $\tau_t^i = (o_0^i, a_0^i, \ldots, o_t^i)$ denotes the agent's action-observation history. The joint action $\mathbf{a}_t = (a_t^1, \ldots, a_t^N)$ induces a transition $s_{t+1} \sim P(\cdot \mid s_t, \mathbf{a}_t)$ and a shared reward $r_t = R(s_t, \mathbf{a}_t)$.

Under CTDE, a centralized value function $Q_{\text{tot}}(s_t, \mathbf{a}_t)$ may be learned during training to facilitate credit assignment, while execution relies solely on decentralized policies $\{\pi_v^i\}_{i=1}^N$. This framework encompasses widely used value-based MARL algorithms such as VDN, QMIX, and QPLEX.

### 3.2. Limited-Budget Action Hijacking Adversary

We study a test-time adversary that interferes with the execution of the victim policies by hijacking agent actions. The adversary does not modify environment dynamics, observations, or rewards. Instead, it selectively replaces the actions of a subset of agents before execution.

Formally, at each timestep $t$, the adversary selects a victim set $\mathcal{V}_t \subseteq \{1, \ldots, N\}$ with cardinality constraint $|\mathcal{V}_t| \leq K$, where $K$ denotes the per-timestep attack capacity. For each hijacked agent $i \in \mathcal{V}_t$, the original action $a_t^i$ is replaced by an adversarial action $a_t^{i,\text{adv}}$, while actions of non-targeted agents remain unchanged.

The adversary is subject to an explicit episode-level attack budget $B$, which limits the number of timesteps at which attacks may occur within each episode. Let $B_t$ denote the remaining budget at timestep $t$, initialized as $B_0 = B$. If $\mathcal{V}_t \neq \emptyset$, one unit of budget is consumed:

$$B_{t+1} = B_t - \mathbb{I}[\mathcal{V}_t \neq \emptyset],$$

and when $B_t = 0$, only the no-op action $\mathcal{V}_t = \emptyset$ is permitted.

The adversary aims to minimize the cumulative return of the victim team, i.e.,

$$\max_{\pi_{\text{adv}}} \mathbb{E}_{\pi_v, \pi_{\text{adv}}} \left[ \sum_{t=0}^T -r_t \right],$$

subject to the per-timestep capacity constraint $K$ and the episode-level budget $B$.

The above formulation induces a Limited Policy Adversary Dec-POMDP (LPA-Dec-POMDP) (Yuan et al., 2023), in which an external adversary intervenes in action execution under explicit resource constraints. Directly solving the optimal adversarial policy in this setting is generally intractable due to the need to jointly reason over attack timing, victim coordination, and action manipulation over long horizons. Accordingly, we adopt a learnable approximation by parameterizing the adversary as a hierarchical policy and optimizing it via policy gradient methods, yielding a practical and scalable solution compatible with CTDE MARL systems.

## 4. Methodology

We propose a hierarchical, budget-efficient adversarial attack method, together with an efficient adversarial training scheme. Our approach comprises (i) a hierarchical attacker that produces budget-efficient multi-agent attacks under explicit budget constraints, and (ii) an alternating adversarial training procedure in which the victim policy is trained against attackers sampled from a pool of previously learned adversaries.

### 4.1. Hierarchical Budget-Efficient Adversarial Attacks

Directly optimizing a flat adversarial policy requires jointly modeling attack timing, victim selection, and action replacement. Optimizing them jointly within a single policy often leads to inefficient budget usage and unstable training. To address this issue, we design a budget-aware hierarchical adversary. The adversarial policy $\pi_{\text{adv}}$ is factorized into two components: a *victim selector* $\pi_{\text{sel}}$ that decides *when* and *which* agents to hijack, and an *action attacker* $\pi_{\text{act}}$ that determines the adversarial actions for selected victims. This decomposition allows the adversary to reason explicitly about attack timing, victim coordination, and action manipulation under limited attack opportunities.

**Observation Encoding.** At each timestep $t$, the adversary receives joint agent observations $\mathbf{o}_t$ and the remaining episode budget $B_t$. To capture coordination structure among agents, we first embed each agent's observation using a lightweight graph-based encoder. Specifically, for agent $i$, we compute

$$\mathbf{h}_t^i = f_{\text{enc}} \left( o_t^i, \; \frac{1}{N} \sum_{j=1}^N o_t^j \right),$$

where $f_{\text{enc}}$ is a shallow neural network that combines individual features with a mean-aggregated global context. This design enables the adversary to reason about relative agent

roles and coordination patterns while remaining computationally efficient.

**Budget-Aware Victim Selection.** We begin with the simplest setting where the per-timestep attack capacity is $K = 1$. At each timestep $t$, the adversary makes a hierarchical decision: *when* to initiate an attack, consuming one unit of episode-level budget, and, conditional on attacking, *which* agent to hijack. Let $a_t \in \{0, 1\}$ denote whether an attack is initiated at timestep $t$, and let $i_t \in \{1, \dots, N\}$ denote the victim identity when $a_t = 1$. The *when* decision is parameterized as

$$\Pr(a_t = 1 \mid \mathbf{h}_t, B_t) \triangleq p_{\text{attack}},$$

$$\Pr(a_t = 0 \mid \mathbf{h}_t, B_t) \triangleq p_{\text{noop}} = 1 - p_{\text{attack}}.$$

where $\mathbf{h}_t$ denotes the observation embedding and $B_t$ the remaining episode budget. Conditional on $a_t = 1$, the *which* decision is governed by a categorical victim-selection policy

$$\pi_{\text{vic}}(i_t \mid \mathbf{h}_t, B_t), \quad i_t \in \{1, \dots, N\}.$$

This *when–which* hierarchy defines the semantic structure of budget-aware victim selection. In practice, the hierarchical policy can be optimized via an equivalent flat $(N+1)$-class parameterization, obtained by augmenting the action space with a no-op outcome noop and defining a single categorical selector

$$\pi_{\text{sel}}(i_t \mid \mathbf{h}_t, B_t), \quad i_t \in \{1, \dots, N, \text{noop}\}.$$

The correspondence between the hierarchical formulation and the $(N+1)$-class selector is

$$\pi_{\text{sel}}(\text{noop} \mid \cdot) \triangleq p_{\text{noop}},$$

$$\pi_{\text{sel}}(i_t \mid \cdot) \triangleq p_{\text{attack}} \pi_{\text{vic}}(i_t \mid \cdot), \quad i_t \in \{1, \dots, N\}.$$

Conversely, given any $(N+1)$-class selector $\pi_{\text{sel}}$, the hierarchical components are recovered by setting $p_{\text{noop}} = \pi_{\text{sel}}(\text{noop} \mid \cdot)$, $p_{\text{attack}} = 1 - p_{\text{noop}}$, and defining

$$\pi_{\text{vic}}(i_t \mid \cdot) \triangleq \frac{\pi_{\text{sel}}(i_t \mid \cdot)}{p_{\text{attack}}}, \quad i_t \in \{1, \dots, N\},$$

obtained by restricting $\pi_{\text{sel}}$ to non-noop outcomes and renormalizing. The hierarchical and $(N+1)$-class representations are therefore strictly equivalent. Throughout this section, we use the hierarchical formulation to describe the decision semantics and the $(N+1)$-class selector for optimization.

Under this representation, the selector log-probability decomposes as

$$\log \pi_{\text{sel}}(i_t) = \log p_{\text{attack}} + \log \pi_{\text{vic}}(i_t), \quad i_t \in \{1, \dots, N\}, \tag{1}$$

and

$$\log \pi_{\text{sel}}(\text{noop}) = \log p_{\text{noop}}. \tag{2}$$

Then we extend the same hierarchical formulation to the more general case with per-timestep attack capacity $K > 1$. The semantic structure of the decision remains unchanged: the adversary first decides *when* to initiate an attack, and, conditional on attacking, decides *which* agents to hijack. The difference from the $K = 1$ case lies solely in the *which* decision, which now allocates multiple victims within a single timestep.

Specifically, conditioned on $a_t = 1$, the *which* decision is defined by a sequence of victim-selection policies

$$\pi_{\text{vic}}^{(k)}\big(i_t^k \mid \mathbf{h}_t, B_t, i_t^{<k}\big), \quad i_t^k \in \mathcal{V}_t^k, \quad k = 1, \dots, K,$$

where

$$\mathcal{V}_t^k = \{1, \dots, N\} \setminus i_t^{<k}, \quad i_t^{<k} = \{i_t^1, \dots, i_t^{k-1}\}.$$

That is, conditioned on initiating an attack, victims are selected sequentially without replacement. These policies generalize the single-victim policy $\pi_{\text{vic}}$ used in the $K = 1$ case and jointly define the hierarchical *which* decision when multiple agents may be hijacked.

To optimize this hierarchical policy, we again adopt an equivalent $(N+1)$-class parameterization, following the same construction as in the $K = 1$ case. Within each timestep, the selector realizes the above constrained victim selection by performing $K$ sequential categorical selections over the augmented action set $\{1, \dots, N, \text{noop}\}$. Specifically, the selector generates an ordered sequence $i_t = (i_t^1, \dots, i_t^K)$, where

$$i_t^k \sim \pi_{\text{sel}}\big(\cdot \mid \mathbf{h}_t, B_t, i_t^{<k}\big), \quad i_t^k \in \mathcal{A}_t^k,$$

and the feasible action set $\mathcal{A}_t^k \subseteq \{1, \dots, N, \text{noop}\}$ is defined as

$$\mathcal{A}_t^k = \left\{ i \in \{1, \dots, N, \text{noop}\} \; \middle| \; \begin{array}{c} i \notin i_t^{<k}, \\ (B_t > 0) \vee (i = \text{noop}), \\ i = \text{noop only if } \forall j < k, \; i_t^j = \text{noop} \end{array} \right\}. \tag{3}$$

For the hierarchical *when–which* policy defined above, the sequence-level selector log-probability can be written in a form aligned with the $K = 1$ case. Using the equivalent autoregressive $(N+1)$-class realization and the induced per-step victim policies, we have

$$\log \pi_{\text{sel}}(i_t) = \log p_{\text{attack}} + \sum_{k=1}^{K} \log \pi_{\text{vic}}^{(k)}\big(i_t^k \mid \mathbf{h}_t, B_t, i_t^{<k}\big), \tag{4}$$

while the all-noop outcome reduces to $\log p_{\text{noop}}$. When $K = 1$, $\mathcal{A}_t^1 = \{1, \dots, N, \text{noop}\}$ and Eq. (4) reduces exactly to Eq. (1).

For Proximal Policy Optimization (PPO) (Schulman et al., 2017), the entire selection outcome is treated as a single selector action, with $\ell_t^{\mathrm{sel}} \triangleq \log \pi_{\mathrm{sel}}(i_t)$. The corresponding PPO importance ratio is then defined as

$$r_t^{\mathrm{sel}}(\theta) = \exp\!\big(\ell_t^{\mathrm{sel}}(\theta) - \ell_{t,\mathrm{old}}^{\mathrm{sel}}\big). \tag{5}$$

**Adversarial Action Policy.** Given a selected victim set $\mathcal{V}_t$, the adversary replaces the original actions of the hijacked agents. For each $i \in \mathcal{V}_t$, an adversarial action is sampled from

$$a_t^{i,\mathrm{adv}} \sim \pi_{\mathrm{act}}(\cdot \mid o_t^i, \mathbf{h}_t^i),$$

where invalid actions are masked according to the environment's action availability constraints. The action attacker $\pi_{\mathrm{act}}$ is implemented as a lightweight policy network that operates independently for each selected victim. It does not require access to the victim policy's internal value functions or gradients, thereby avoiding reliance on white-box victim models and improving generality.

For PPO training, we define the action-attacker log-probability at timestep $t$ as the sum of log-probabilities over all hijacked agents,

$$\ell_t^{\mathrm{act}} = \sum_{i \in \mathcal{V}_t} \log \pi_{\mathrm{act}}\!\left(a_t^{i,\mathrm{adv}} \mid o_t^i, \mathbf{h}_t^i\right), \tag{6}$$

where action availability constraints are applied to mask invalid actions. The corresponding PPO importance ratio is

$$r_t^{\mathrm{act}}(\theta) = \exp\!\big(\ell_t^{\mathrm{act}}(\theta) - \ell_{t,\mathrm{old}}^{\mathrm{act}}\big), \tag{7}$$

and this term is applied only at timesteps where $\mathcal{V}_t \neq \emptyset$.

The complete adversarial policy therefore factorizes as

$$\pi_{\mathrm{adv}} = \pi_{\mathrm{sel}} \times \pi_{\mathrm{act}},$$

where $\pi_{\mathrm{sel}}$ determines the victim set $\mathcal{V}_t$ (or a no-op), and $\pi_{\mathrm{act}}$ specifies adversarial actions for the selected agents. This hierarchical structure explicitly separates high-level decisions on *when* and *which* agents to attack from low-level action manipulation, enabling coordinated multi-agent hijacking under a limited attack budget.

**Optimizing the Hierarchical Adversary.** We optimize both the selector policy $\pi_{\mathrm{sel}}$ and the action-attacker policy $\pi_{\mathrm{act}}$ using PPO with the clipped surrogate objective. Let $\hat{A}_t^{\mathrm{sel}}$ and $\hat{A}_t^{\mathrm{act}}$ denote generalized advantage estimates (GAE) computed from the adversarial reward $-r_t$ for the selector and the action attacker, respectively. Let $r_t^x(\theta_x)$ denote the PPO importance ratio, where $x \in \{\mathrm{sel}, \mathrm{act}\}$, with Eq. (5) used for the selector and Eq. (7) for the action attacker. We maximize the clipped objective

$$\mathcal{L}_x^{\mathrm{CLIP}}(\theta_x) = \mathbb{E}_t\left[\min\!\Big(r_t^x(\theta_x)\hat{A}_t^x,\ \mathrm{clip}\big(r_t^x(\theta_x),\, 1{\pm}\epsilon\big)\hat{A}_t^x\Big)\right], \tag{8}$$

where $x \in \{\mathrm{sel}, \mathrm{act}\}$. In addition, we employ standard value-function regression and entropy regularization for each policy. The complete rollout and update procedure is summarized in Appendix A.

---

**Algorithm 1** Alternating Adversarial Training with Budget-Efficient Attacks

---

1: **Initialize:** Victim policy $\pi_v$, Adversary policy $\pi_{\mathrm{adv}}$, Policy pool $\mathcal{P} \leftarrow \{\pi_{\mathrm{adv}}\}$
2: **for** iteration $r = 1, 2, \ldots, R$ **do**
3:     *// Phase 1: Evolve Adversary (Attack)*
4:     Update $\pi_{\mathrm{adv}}$ against fixed $\pi_v$ for $E_{\mathrm{adv}}$ episodes
5:     **Snapshot** current adversary: $\pi_{\mathrm{snap}} \leftarrow \pi_{\mathrm{adv}}$
6:     **Update pool**: $\mathcal{P} \leftarrow \mathcal{P} \cup \{\pi_{\mathrm{snap}}\}$
7:     *// Phase 2: Robustify Victim (Defend)*
8:     **for** episode $e = 1, \ldots, E_{\mathrm{vic}}$ **do**
9:         Sample adversary $m \sim p(m)$
10:         Rollout episode with $\pi_v$ vs $\pi_{\mathrm{adv}}^{(m)}$
11:         Update $\pi_v$ using collected experience
12:     **end for**
13: **end for**

---

### 4.2. Adversarial Training with Budget-Efficient Attacks

We employ an alternating adversarial training framework, termed BHEA-AT, that iteratively updates a victim policy $\pi_v$ and an adversary policy $\pi_{\mathrm{adv}}$. At each iteration, the adversary is optimized against a fixed victim, followed by training the victim to improve robustness. The procedure is summarized in Algorithm 1.

After each adversary update round, the current adversary is snapshotted and added to a policy pool

$$\mathcal{P}_t = \{\pi_{\mathrm{adv}}^{(1)}, \ldots, \pi_{\mathrm{adv}}^{(M_t)}\},$$

where $M_t = |\mathcal{P}_t|$ denotes the current pool size and larger indices correspond to more recent snapshots. The pool size is capped at $M_{\mathrm{max}}$ by discarding the oldest adversary when necessary. During data collection, a single adversary is sampled from the pool at the beginning of each episode and kept fixed throughout the episode.

To mildly bias training toward stronger adversaries while retaining diversity, we adopt a simple rank-based recency sampling scheme. Adversaries are ordered chronologically, and the sampling probability of $\pi_{\mathrm{adv}}^{(m)}$ is defined as

$$p_t(m) = \frac{m}{\sum_{j=1}^{M_t} j} = \frac{2m}{M_t(M_t + 1)}, \qquad m = 1, \ldots, M_t. \tag{9}$$

This scheme favors more recent adversaries while ensuring non-zero sampling probability for all snapshots, introduces no additional tuning parameters, and naturally adapts to the growing pool size in early training.

As shown in Algorithm 1, adversary evolution and victim training alternate across outer iterations. In the victim update phase, adversaries are sampled from the pool according to Eq. (9), exposing the victim to a progressively stronger yet diverse set of attack behaviors.

# 5. Experiments

We evaluate BHEA and BHEA-AT on SMAC across diverse coordination scenarios, comparing against representative adversarial training baselines under matched attack budgets, with a focus on robustness, generalization to unseen attacks, and training efficiency. Additional results on MPE benchmarks and policy-gradient victims (MAPPO and MADDPG) are reported in Appendix H.

## 5.1. Performance Comparison in SMAC

We compare robust MARL training strategies on the SMAC benchmark under diverse attack settings. Table 1 reports the average test win rate (%) across six representative scenarios with varying coordination complexity. All methods are evaluated under matched attack budgets for fair comparison, with experimental details provided in Appendix C.

Across all scenarios, robust training does not introduce noticeable performance degradation, and BHEA-AT matches or slightly outperforms the baselines under natural execution. Performance differences become more evident under strong adaptive attacks, including EGA and Wolfpack. Vanilla QMIX suffers substantial degradation, particularly on coordination-intensive scenarios such as 8m and MMM. ROMANCE and WALL mitigate these failures to some extent but exhibit considerable performance variation across tasks. In contrast, BHEA-AT maintains consistently high win rates across all scenarios under both EGA and Wolfpack attacks. The improvements are most pronounced on complex maps, where manipulating agent actions at critical timesteps is especially damaging. These results indicate that our method more effectively addresses the coordination vulnerabilities targeted by adaptive adversaries.

We further evaluate all methods under BHEA, which is specifically designed to target critical coordination failures under a limited attack budget and thus serves as a stringent stress test for robustness. Under this setting, existing methods experience large performance drops on several scenarios. By comparison, BHEA-AT preserves high win rates across all tasks, including the most challenging scenarios such as MMM and 8m. Overall, these results demonstrate that BHEA-AT achieves stronger and more consistent robustness under targeted attacks, while maintaining competitive performance under natural execution.

While the above results summarize final robustness under matched attack budgets, robustness in practice also requires generalization to adversarial behaviors not used during adversarial training. To examine this aspect, we evaluate all methods against unseen attacks, where the victim policies are tested under attack instances that are never used to generate training data during adversarial training. Figure 1 reports the evolution of test win rates under unseen attacks for victim policies trained with three different MARL learners (QMIX, VDN, and QPLEX) on two representative scenarios, the 1c3s5z map and the MMM map.

Across all victim learning algorithms and both scenarios, the Vanilla and RANDOM baselines exhibit consistently low win rates under unseen attacks, indicating limited generalization. ROMANCE and WALL achieve partial improvements but show noticeable performance fluctuations during training. In contrast, BHEA-AT rapidly attains high win rates early in training and maintains stable performance thereafter. This trend is consistent across victim algorithms and is especially pronounced on the challenging MMM scenario, suggesting that the robustness gains of our method do not arise from overfitting to a particular training adversary, but instead reflect an improved ability to preserve coordination under limited-step action hijacking, even when adversarial behaviors differ from those encountered during training.

## 5.2. Budget Efficiency and Ablation Analysis

**Sensitivity to Attack Budget and Capacity.** We further analyze the effectiveness of BHEA by systematically varying the episode-level attack budget $B$ and the per-timestep attack capacity $K$.

We first fix the episode-level budget $B$ and vary the per-timestep attack capacity $K$. The two upper plots of Figure 2 show that increasing $K$ generally leads to stronger attacks, as the adversary can manipulate multiple agents within the same timestep and induce more coordinated disruption. We next fix the per-timestep capacity to $K = 1$ and vary the episode-level attack budget $B$. As shown in the two lower plots of Figure 2, larger $B$ consistently results in lower win rates, reflecting the increased opportunities for adversarial intervention.

Overall, these results indicate that BHEA scales effectively with both budget dimensions. Even under limited budgets, BHEA is able to exploit critical timesteps and agent coordination to induce substantial performance degradation. This behavior is consistent with the design of BHEA, which explicitly reasons about attack timing and coordinated victim selection under explicit resource constraints.

**Ablation Study.** We conduct ablation experiments to examine the contribution of key components in BHEA and their impact when used for adversarial training. We consider three variants: –Selector, where victim agents are selected uniformly at random; –Attacker, where the actions

Table 1. Average test win rate (%) under different attack settings on SMAC. Results are reported as mean ± standard deviation.

| Attack Setting | Scenario / Method | 2s3z | 3m | 3s_vs_3z | 8m | MMM | 1c3s5z |
|---|---|---|---|---|---|---|---|
| Natural | Vanilla QMIX | 98.1±1.4 | 99.0±0.6 | 99.1±0.2 | 97.2±1.5 | 99.0±0.8 | 99.2±0.6 |
| | RANDOM | 98.0±1.1 | 98.2±1.3 | 99.0±0.9 | 98.1±1.0 | 98.3±1.1 | 99.1±0.7 |
| | ROMANCE | 95.4±1.1 | 98.1±1.0 | 99.2±0.5 | 96.3±1.8 | 98.1±1.2 | 98.4±1.0 |
| | WALL | 98.2±1.0 | 99.1±0.6 | 99.6±0.2 | 99.0±0.7 | 99.2±0.5 | 98.3±0.9 |
| | **BHEA-AT** | **98.5±0.8** | **99.3±0.4** | **99.7±0.2** | **99.4±0.2** | **99.4±0.4** | **99.4±0.4** |
| Random Attack | Vanilla QMIX | 90.2±3.4 | 88.1±4.1 | 93.3±2.5 | 96.1±1.9 | 81.4±5.2 | 99.1±0.6 |
| | RANDOM | 95.1±2.0 | 94.2±2.3 | 96.1±1.8 | 96.2±1.5 | 90.3±3.1 | 99.0±0.7 |
| | ROMANCE | 96.3±1.5 | 96.1±1.6 | 98.2±1.1 | 97.1±1.2 | 94.2±2.0 | 98.3±1.0 |
| | WALL | 97.1±1.2 | 96.3±1.4 | 99.1±0.6 | 96.4±1.4 | 93.1±2.5 | 99.2±0.5 |
| | **BHEA-AT** | **97.4±0.9** | **97.2±1.1** | **99.3±0.5** | **98.1±0.8** | **98.4±1.0** | **99.3±0.4** |
| EGA | Vanilla QMIX | 88.3±4.5 | 78.1±5.2 | 71.4±6.1 | 56.2±7.3 | 53.1±8.0 | 90.2±3.5 |
| | RANDOM | 88.1±4.2 | 82.3±4.8 | 75.2±5.5 | 57.4±6.9 | 55.3±7.5 | 92.1±3.1 |
| | ROMANCE | 94.2±2.1 | 85.1±3.5 | 98.1±1.2 | 63.2±6.1 | 92.1±2.8 | 94.3±2.2 |
| | WALL | 95.1±1.8 | 81.2±4.0 | 97.3±1.5 | 77.1±5.2 | 91.2±3.0 | 98.1±1.1 |
| | **BHEA-AT** | **95.3±1.5** | **90.4±2.8** | **98.4±1.0** | **78.3±4.5** | **98.1±1.2** | **98.4±0.9** |
| Wolfpack Attack | Vanilla QMIX | 76.2±5.1 | 73.1±6.0 | 47.3±8.2 | 68.1±6.5 | 41.2±9.1 | 94.1±2.5 |
| | RANDOM | 77.1±4.8 | 78.3±5.5 | 54.1±7.8 | 65.2±6.8 | 74.3±5.2 | 94.2±2.4 |
| | ROMANCE | 94.2±2.3 | 85.4±3.2 | 95.1±2.1 | 81.3±4.1 | 91.2±3.0 | 95.3±1.8 |
| | WALL | 94.1±2.0 | 81.2±3.8 | 99.1±0.7 | 89.2±3.5 | 92.1±2.8 | 96.1±1.5 |
| | **BHEA-AT** | **96.2±1.4** | **91.3±2.5** | **99.2±0.6** | **92.1±2.9** | **98.3±1.1** | **99.1±0.8** |
| BHEA | Vanilla QMIX | 10.2±3.5 | 12.1±4.1 | 3.2±1.8 | 11.3±2.5 | 8.1±3.2 | 22.1±5.5 |
| | RANDOM | 14.3±4.2 | 23.1±5.5 | 15.2±4.8 | 17.1±3.2 | 16.2±4.5 | 21.3±5.1 |
| | ROMANCE | 85.1±3.8 | 36.2±6.2 | 65.3±7.1 | 45.1±6.8 | 80.2±4.5 | 40.1±6.5 |
| | WALL | 45.2±6.5 | 68.3±5.8 | 47.1±7.5 | 69.2±5.9 | 63.1±6.0 | 26.2±5.8 |
| | **BHEA-AT** | **95.1±1.6** | **75.2±4.1** | **99.5±0.4** | **93.1±2.5** | **99.2±0.8** | **87.3±3.5** |

Table 2. Training time (hours) across scenarios.

| Scenario / Method | 2s3z | 3m | 3s_vs_3z | 8m | MMM | 1c3s5z |
|---|---|---|---|---|---|---|
| Vanilla QMIX | 8h | 8h | 9h | 8h | 9h | 9h |
| Random | 8h | 8h | 9h | 8h | 9h | 9h |
| Romance | 25h | 26h | 30h | 28h | 30h | 30h |
| Wall | 43h | 45h | 47h | 45h | 47h | 46h |
| BHEA-AT | 16h | 16h | 18h | 17h | 18h | 18h |

of selected victims are replaced with random actions; and –Attacker + WorstQ, where the actions of selected victims are replaced by the lowest-Q action according to the victim policy.

We first analyze how different attacker ablations affect adversarial training. The two upper plots of Figure 3 show the performance of victim policies trained against ablated attackers on MMM and 1c3s5z. Replacing the learned victim selector (–Selector) or the learned action attacker (–Attacker) leads to weaker robustness. Using the worst-Q action for substitution (–Attacker + WorstQ) improves over random replacement, but still underperforms the full attacker. We next examine the effect of these ablations on the attack itself. The lower plot of Figure 3 shows that the full BHEA consistently induces the strongest performance degradation,

while ablated variants are less effective.

Overall, these results indicate that the effectiveness of BHEA, both as an attack and as a training opponent, relies on the combination of budget-aware victim selection and learned action manipulation, rather than any single heuristic component. To better understand the necessity of hierarchical attack parameterization, we include an additional empirical study comparing against a flat adversarial policy in Appendix D.

### 5.3. Efficiency and Robustness in Practice

We evaluate BHEA-AT from a practical perspective, with an emphasis on training efficiency and robustness beyond standard benchmark settings. Experimental details are provided in Appendix C.

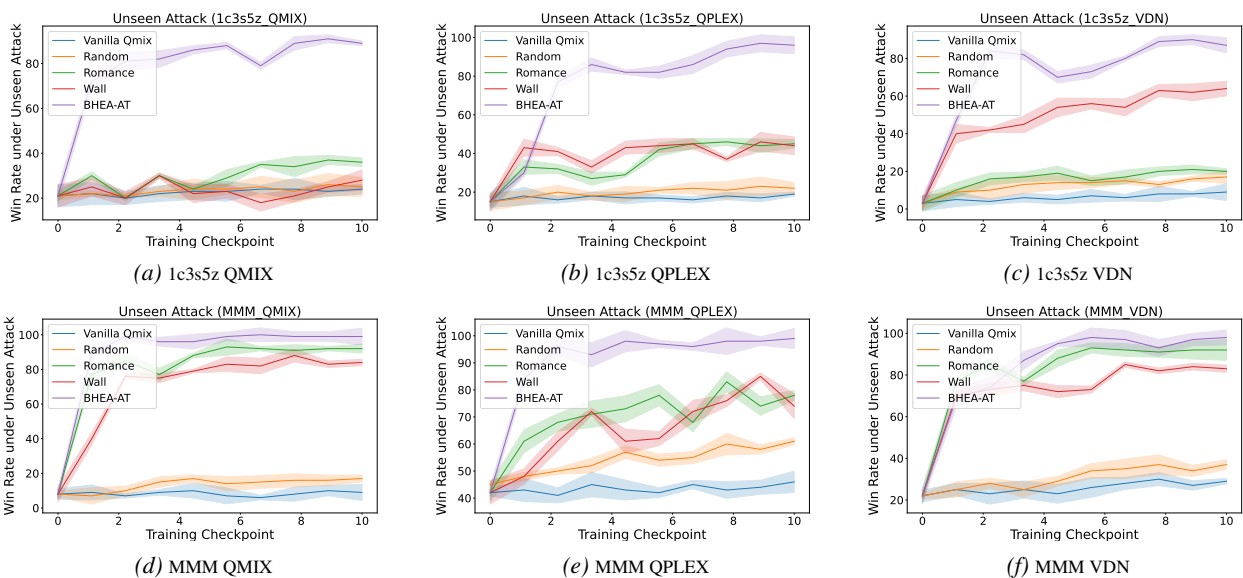

*Figure 1.* Victim win rate under unseen attacks as a function of training episodes. The x-axis denotes training episodes, and the y-axis reports the victim win rate under attack. Results are shown for victim policies trained with three different MARL learners (QMIX, VDN, and QPLEX) on two representative SMAC scenarios, 1c3s5z and MMM.

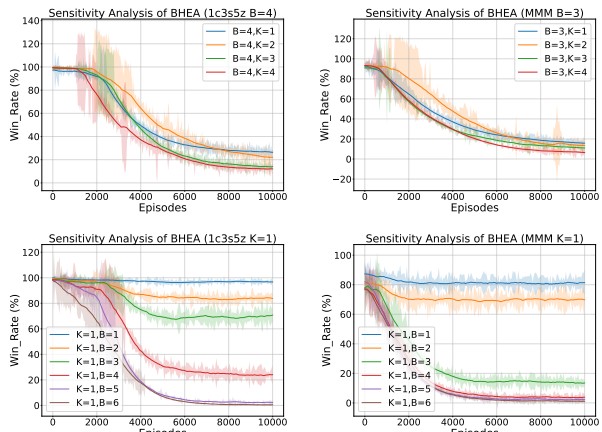

*Figure 2.* Sensitivity analysis of adversarial attacks under varying budget constraints (B, K)

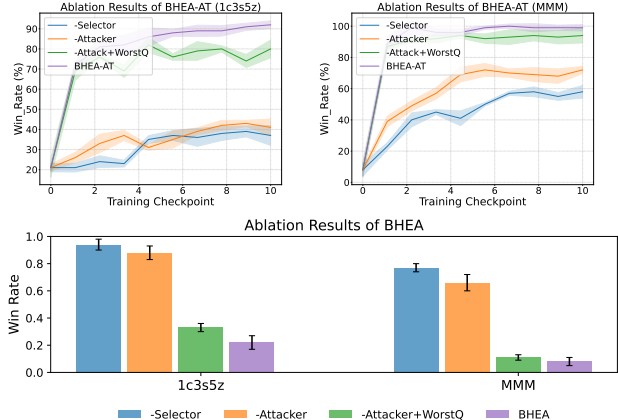

*Figure 3.* Ablation results of BHEA and BHEA-AT. The two upper plots show the performance of victim policies trained with ablated attackers on 1c3s5z and MMM. The lower plot shows the attack effectiveness of ablated variants of BHEA.

We compare the computational cost of different adversarial training approaches. Table 2 reports the training time across six SMAC scenarios. Across all scenarios, BHEA-AT is substantially more computationally efficient than planner-based and population-based adversarial training methods such as WALL and ROMANCE. In particular, BHEA-AT reduces training time by approximately 35–40% compared to ROMANCE, and by more than 60% compared to WALL, while achieving stronger robustness. To provide a more comprehensive comparison beyond wall-clock time, we further report profiler-estimated training FLOPs and peak GPU memory usage in Appendix F.

While Vanilla and Random baselines incur similar train-

ing costs, they provide limited robustness under adversarial attacks. By contrast, BHEA-AT achieves strong and consistent robustness with significantly lower training cost, owing to its stronger hierarchical adversary and the avoidance of computationally expensive planning or population search.

We further evaluate robustness under degraded initial conditions, where agents start from a disadvantaged state. Detailed results are reported in Appendix G.

## 6. Limitations

Our threat model assumes a strong execution-time attacker who has black-box access to the victim policy, observes team-level execution information, and can replace the executed actions of selected agents within a given budget. Although this stronger attack assumption is useful for stress-testing cooperative MARL policies, it goes beyond purely local or compromised-agent-only attack scenarios. To provide a more balanced evaluation, we also include restricted-access experiments under weaker-access settings in Appendix E. In addition, our experiments are limited to SMAC and MPE benchmarks with relatively small teams, leaving scalability to larger multi-agent systems for future study. Finally, although these benchmarks capture partial observability and cooperative coordination, they remain simulation-based and do not fully reflect the noise, physical constraints, and safety requirements of real-world deployments. Evaluating robustness under larger, adaptive, and more realistic attack budgets is an important direction for future work.

## 7. Conclusion

We proposed BHEA, a budget-aware hierarchical adversarial attack for cooperative MARL, together with BHEA-AT, an efficient adversarial training framework built upon it. By explicitly reasoning about attack timing and coordinated victim selection, our approach enables strong attacks under limited budgets and provides effective training signals for robustness. Experiments demonstrate improved robustness and training efficiency compared to existing adversarial training methods.

## Impact Statement

This paper presents work whose goal is to advance the field of Machine Learning. There are many potential societal consequences of our work, none which we feel must be specifically highlighted here.

## Acknowledgment

This work is supported in part by New Generation Artificial Intelligence-National Science and Technology Major Project (2025ZD0123504), in part by the National Natural Science Foundation of China under Grant No.62576100 and No.62203113, and in part by the Big Data Computing Center of Southeast University.

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

# A. Implementation Details of the Hierarchical Adversary

This appendix provides implementation details of the proposed hierarchical adversary and the alternating adversarial training procedure introduced in Section 4. It complements Algorithms 2 and 1 by specifying network parameterization, constraint enforcement, and practical training details.

## A.1. Network Architecture and Parameterization

The adversarial policy $\pi_{\mathrm{adv}}$ consists of a victim selector $\pi_{\mathrm{sel}}$ and an action attacker $\pi_{\mathrm{act}}$, each equipped with an independent value function for PPO optimization.

**Observation encoder.** Each agent observation $o_t^i$ is embedded using a lightweight graph-based encoder. In practice, this encoder is implemented as a shallow MLP that combines the individual observation with a mean-aggregated global context:

$$\mathbf{h}_t^i = f_{\mathrm{enc}}\left(o_t^i, \ \frac{1}{N}\sum_{j=1}^{N} o_t^j\right).$$

The encoder parameters are shared across agents and timesteps. A global adversarial embedding $\mathbf{h}_t$ is obtained by averaging $\{\mathbf{h}_t^i\}_{i=1}^{N}$.

**Selector and attacker networks.** The selector $\pi_{\mathrm{sel}}$ is implemented as an MLP that takes $(\mathbf{h}_t, B_t)$ as input and outputs logits over agents and the no-op action, with action masking applied to enforce feasibility constraints. For $K > 1$, the same network is reused autoregressively with updated masks.

The action attacker $\pi_{\mathrm{act}}$ is a lightweight MLP applied independently to each selected victim, taking $(o_t^i, \mathbf{h}_t^i)$ as input. Invalid actions are masked according to environment-provided action availability.

Each policy head is paired with its own value network, denoted by $V_{\mathrm{sel}}(\mathbf{h}_t, B_t)$ and $V_{\mathrm{act}}(\mathbf{h}_t, B_t)$, respectively.

## A.2. Budget and Capacity Constraint Enforcement

Budget and capacity constraints are enforced via action masking rather than penalty terms. When the remaining budget $B_t = 0$, all selector logits except the no-op action are masked. For $K > 1$, autoregressive selection masks ensure that: (i) agents are not selected multiple times, (ii) no-op is only allowed before any agent is selected, and (iii) once an attack is initiated, remaining capacity is force-filled unless exhausted.

Budget consumption is implemented at the timestep level: any non-empty victim set $\mathcal{V}_t$ consumes exactly one unit of budget, independent of $|\mathcal{V}_t|$.

## A.3. Trajectory Collection and Advantage Estimation

During adversary training, the victim policy $\pi_v$ is fixed. At each timestep, the adversary selects a victim set $\mathcal{V}_t$, optionally overrides victim actions, and receives the negated environment reward $-r_t$.

Each stored transition includes:
$$(\mathbf{h}_t, B_t, \mathcal{V}_t, \ell_t^{\mathrm{sel}}, \ell_t^{\mathrm{act}}, r_t, V_t^{\mathrm{sel}}, V_t^{\mathrm{act}}).$$

Generalized Advantage Estimation (GAE) is applied separately for the selector and the action attacker using their corresponding value predictions. Both policies share the same return targets $\hat{R}_t$, while advantages $\hat{A}_t^{\mathrm{sel}}$ and $\hat{A}_t^{\mathrm{act}}$ are computed with respect to $V_{\mathrm{sel}}$ and $V_{\mathrm{act}}$, respectively, using discount factor $\gamma$ and trace parameter $\lambda$.

## A.4. PPO Optimization Details

Selector and action attacker policies are optimized jointly using PPO. Separate importance sampling ratios are computed for $\pi_{\mathrm{sel}}$ and $\pi_{\mathrm{act}}$ as defined in Eqs. (5) and (7). The final loss is the sum of clipped surrogate objectives for the selector and the action attacker, augmented with independent value regression and entropy regularization:

$$\mathcal{L} = \mathcal{L}_{\mathrm{sel}}^{\mathrm{CLIP}} + \mathcal{L}_{\mathrm{act}}^{\mathrm{CLIP}} + \alpha\|V_{\mathrm{sel}} - \hat{R}_t\|^2 + \alpha\|V_{\mathrm{act}} - \hat{R}_t\|^2 - \beta H. \tag{10}$$

All adversarial parameters are updated end-to-end via backpropagation.

## A.5. Practical Training Hyperparameters

Unless otherwise specified, we employ standard PPO hyperparameters and optimize the adversarial policy using the Adam optimizer. Detailed hyperparameter settings are summarized in Table 3.

*Table 3.* Hyperparameters for Adversarial Training.

| Parameter | Value |
|---|---|
| Optimizer | Adam |
| Learning Rate | $3 \times 10^{-4}$ |
| Discount Factor ($\gamma$) | 0.99 |
| GAE Parameter ($\lambda$) | 0.95 |
| Clipping Threshold ($\epsilon$) | 0.2 |
| Entropy Coefficient ($\beta$) | 0.05 |
| Value Coefficient ($\alpha$) | 0.5 |
| Rollout Horizon | 2048 |
| Minibatch Size | 512 |
| Optimization Epochs | 4 |

During the alternating adversarial training process, the adversary undergoes $U_{\mathrm{adv}}$ PPO updates per outer iteration, following Algorithm 1.

## B. Additional Background and Discussion

### B.1. Wolfpack and EGA

Wolfpack (Lee et al., 2025) is a planner-based adversarial attack framework for CMARL under the CTDE paradigm. It constructs coordinated attacks by explicitly selecting critical timesteps and groups of victim agents, typically leveraging centralized value functions and, in some cases, learned environment models to estimate the long-term impact of attacks. The attack process is decomposed into procedural components, including step selection, victim group identification, and action replacement.

Evolutionary Generation of Attackers (EGA) (Yuan et al., 2023) is a complementary approach that constructs a population of adversarial policies through evolutionary or population-based search. By maintaining diversity among attackers, EGA exposes the victim policy to a wide range of adversarial behaviors during training, thereby improving robustness.

### B.2. WALL and ROMANCE

WALL (Lee et al., 2025) extends the Wolfpack framework into a robust MARL training paradigm by optimizing victim policies against coordinated adversarial attacks generated by the Wolfpack planner. By explicitly incorporating such attacks during training, WALL encourages agents to develop system-level cooperation strategies that are resilient to targeted multi-agent disruptions.

ROMANCE (Yuan et al., 2023) is a population-based adversarial training framework that improves robustness by sampling attackers from a pool of diverse adversarial policies. Training against a mixture of adversarial behaviors mitigates overfitting to specific attack patterns and has demonstrated strong empirical performance in CMARL benchmarks.

### B.3. Discussion: Relation to Our Framework

Our framework occupies a distinct point in the design space of adversarial MARL. Rather than constructing attacks through explicit planning or population heuristics, we model a resource-constrained adversary as a single hierarchical policy optimized within an LPA-Dec-POMDP.

We now clarify how this modeling choice differs from the above approaches in terms of assumptions and design choices.

---

**Algorithm 2** Hierarchical Budget-Efficient Adversarial Attack

---

**Require:** Victim policy $\pi_v$, Adversarial policy $\pi_{\mathrm{adv}}$ (parameterized by $\theta$), Budget $B$, Capacity $K$.

1: Initialize adversarial parameters $\theta$.
2: **for** iteration $iter = 1, 2, \ldots$ **do**
3:     Initialize episode budget $B_0 \leftarrow B$, replay buffer $\mathcal{D} \leftarrow \emptyset$.
4:     **for** timestep $t = 0, 1, \ldots, T$ **do**
5:         Get observations $\mathbf{o}_t$, compute embeddings $\mathbf{h}_t$.
6:         Initialize victim set $\mathcal{V}_t \leftarrow \emptyset$.
7:         // Phase 1: Hierarchical Victim Selection
8:         **if** $B_t > 0$ **then**
9:             Initialize $\mathcal{V}_t \leftarrow \emptyset$.
10:            **for** $k = 1$ to $K$ **do**
11:                Construct feasible set $\mathcal{A}_t^k$ via Eq. (3).
12:                Sample $i_t^k \sim \pi_{\mathrm{sel}}(\cdot \mid \mathbf{h}_t, B_t, i_t^{<k})$.
13:                Accumulate selector log-prob via Eq. (4).
14:                **if** $i_t^k = \mathrm{noop}$ **then**
15:                    **break**
16:                **else**
17:                    $\mathcal{V}_t \leftarrow \mathcal{V}_t \cup \{i_t^k\}$.
18:                **end if**
19:            **end for**
20:         **end if**
21:         // Phase 2: Action Replacement & Budget Update
22:         **for** each victim $i \in \mathcal{V}_t$ **do**
23:            Sample adversarial action $a_t^{i,\mathrm{adv}} \sim \pi_{\mathrm{act}}(\cdot \mid o_t^i, \mathbf{h}_t^i)$.
24:            Override action $a_t^i \leftarrow a_t^{i,\mathrm{adv}}$.
25:         **end for**
26:         Update budget: $B_{t+1} \leftarrow B_t - \mathbb{I}[\mathcal{V}_t \neq \emptyset]$.
27:         Execute $\mathbf{a}_t$, get reward $-r_t$ (adversarial reward).
28:         Store transition in $\mathcal{D}$.
29:     **end for**
30:     // Phase 3: Adversarial Optimization (PPO)
31:     Compute returns $\{\hat{R}_t\}$ and advantages $\{\hat{A}_t^{\mathrm{sel}}, \hat{A}_t^{\mathrm{act}}\}$ using GAE with value estimates.
32:     Recompute $\ell_t^{\mathrm{sel}}(\theta)$ and $\ell_t^{\mathrm{act}}(\theta)$ (Eqs. (4), (6)).
33:     Form ratios $r_t^{\mathrm{sel}}(\theta)$ and $r_t^{\mathrm{act}}(\theta)$ (Eqs. (5), (7)).
34:     Update $\pi_{\mathrm{sel}}, \pi_{\mathrm{act}}, V_{\mathrm{sel}}, V_{\mathrm{act}}$ by minimizing $\mathcal{L}$ defined in Eq. (10).
35: **end for**

---

**Relation to Wolfpack and WALL.** Wolfpack attacks rely on explicit planning mechanisms to identify critical timesteps and coordinated victim groups, often leveraging centralized value functions or learned environment models. While effective at constructing targeted attacks, the resulting attack logic is largely procedural and closely tied to specific CTDE implementations.

By contrast, our adversary is formulated as a fully learnable policy optimized end-to-end via reinforcement learning. Attack timing and victim coordination emerge from interaction under explicit budget constraints, without explicit planning or white-box access to the victim's value functions. This formulation enables broader compatibility with different MARL backbones and training pipelines.

**Relation to EGA and ROMANCE.** Population-based approaches such as EGA and ROMANCE improve robustness by training against a diverse set of attackers, typically generated via evolutionary or population-based search. While effective, these methods do not explicitly optimize attack timing and victim coordination under budget constraints within a single adversarial policy, and instead rely on population diversity to induce robustness.

Our approach instead focuses on learning a single, budget-aware adversarial policy that explicitly reasons about attack

timing, victim selection, and action replacement under hard constraints. This design enables strong attacks even under low budgets and is complementary to population-based training schemes.

## C. Experimental Setup on SMAC

This appendix describes the SMAC environments and the attack configurations used in our experiments. All experiments in this paper are conducted on a GPU server equipped with an NVIDIA GeForce RTX 3090 GPU. We run each experiment with five different random seeds and report the mean and standard deviation across seeds.

### C.1. SMAC Environments

We conduct experiments on a subset of scenarios from the SMAC benchmark (Samvelyan et al., 2019). SMAC consists of cooperative multi-agent tasks under centralized training and decentralized execution, where agents must coordinate to defeat an opposing team controlled by the built-in StarCraft II AI.

Following prior robustness and adversarial training studies such as Wolfpack (Lee et al., 2025), we select scenarios that vary in team size, unit heterogeneity, and coordination difficulty. The selected maps include 3m, 8m, 2s3z, 3s_vs_3z, MMM, and 1c3s5z, which together cover both homogeneous and heterogeneous unit compositions.

Table 4 summarizes the unit composition and the dimensions of the global state space, local observations, and discrete action space for each scenario. These specifications follow the standard SMAC implementation and are reported for completeness.

*Table 4.* Specifications of SMAC environments used in our experiments.

| Map | Ally Units | Enemy Units | State Dim. | Obs Dim. | Actions |
|-----|-----------|-------------|------------|----------|---------|
| 3m | 3 Marines | 3 Marines | 48 | 30 | 9 |
| 3s_vs_3z | 3 Stalkers | 3 Zealots | 54 | 36 | 9 |
| 2s3z | 2 Stalkers, 3 Zealots | 2 Stalkers, 3 Zealots | 120 | 80 | 11 |
| 8m | 8 Marines | 8 Marines | 168 | 80 | 14 |
| 1c3s5z | 1 Colossus, 3 Stalkers, 5 Zealots | 1 Colossus, 3 Stalkers, 5 Zealots | 270 | 162 | 15 |
| MMM | 1 Medivac, 2 Marauders, 7 Marines | 1 Medivac, 2 Marauders, 7 Marines | 290 | 160 | 16 |

The selected scenarios exhibit diverse coordination characteristics. In particular, 3m and 8m involve homogeneous allied units and require relatively simple coordination, whereas MMM and 1c3s5z feature heterogeneous unit types and demand more complex role assignment and temporal coordination. The 2s3z and 3s_vs_3z maps represent intermediate settings with asymmetric unit capabilities.

### C.2. Attack Configuration

We evaluate robustness under a limited-budget action hijacking adversary. At each timestep, the adversary may override the actions of up to $K$ agents, referred to as the per-timestep attack capacity. In addition, the adversary is constrained by an episode-level attack budget $B$, which limits the total number of timesteps during which attacks may occur.

An attack at timestep $t$ consumes one unit of budget whenever at least one agent is hijacked, regardless of the number of agents affected. Once the budget is exhausted, the adversary is restricted to a no-op action for the remainder of the episode.

Consistent with Wolfpack (Lee et al., 2025), we adopt scenario-dependent values of $(K, B)$ to account for differences in team size and coordination structure across environments. This avoids trivially weak or overly destructive attack settings and ensures comparable attack strength across scenarios.

The attack capacity $K$ and episode-level budget $B$ used for each SMAC scenario are reported in Table 5.

Table 5 summarizes the attack configuration used in Section 5.1. In this comparison, we fix the attack capacity to $K = 1$, as ROMANCE (Yuan et al., 2023) only supports single-agent action hijacking per timestep, and use scenario-dependent episode budgets $B$. For both baselines, all method-specific hyperparameters (e.g., those not shared with BHEA) are set to the default values reported in their respective original papers to ensure a fair comparison. In Section 5.2, we further analyze the effect of varying both the attack capacity $K$ and the episode-level budget $B$ on BHEA.

*Table 5.* Attack capacity $K$ and episode-level attack budget $B$ for each SMAC scenario.

| Scenario | Attack Capacity $K$ | Attack Budget $B$ |
|---|---|---|
| 2s3z | 1 | 4 |
| 3m | 1 | 3 |
| 3s_vs_3z | 1 | 8 |
| 8m | 1 | 8 |
| MMM | 1 | 4 |
| 1c3s5z | 1 | 4 |

## C.3. Adversarial Training

In adversarial training, BHEA and all baseline methods are trained for the same total number of episodes. Specifically, for the 2s3z, 1c3s5z, MMM, and 3s_vs_3z scenarios, each method is trained for 100,000 episodes, while for 3m and 8m, the total training budget is 200,000 episodes.

For BHEA-AT, the attacker training accounts for 30% of the total training episodes and is conducted over 10 alternating rounds. In each round, the attacker is trained for 3,000 episodes (for 100k-episode scenarios) or 6,000 episodes (for 200k-episode scenarios), followed by 7,000 or 14,000 episodes of adversarial training, respectively.

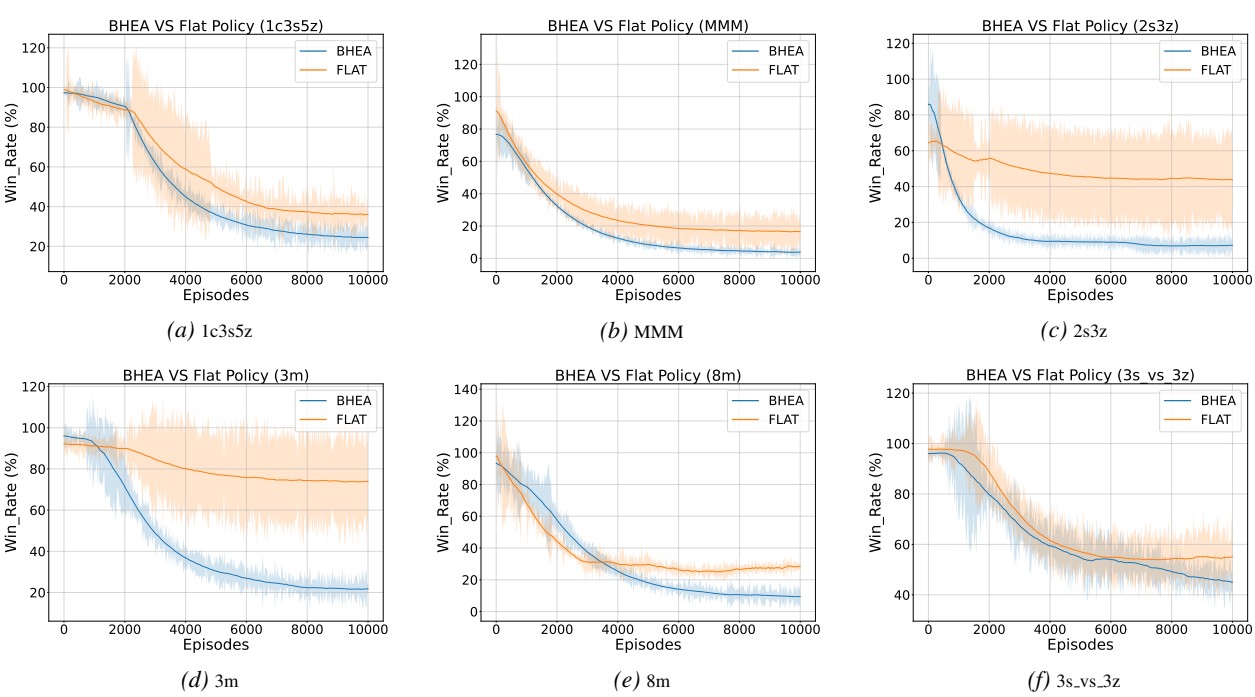

*Figure 4.* Comparison between flat and hierarchical (BHEA) adversarial policies under the $K = 1$ setting. The x-axis denotes training episodes, and the y-axis reports the victim win rate under attack. Results are shown on representative SMAC scenarios with identical victim policies, attack budgets, and training configurations. Although the flat policy has equivalent expressive power to the hierarchical design for single-agent attacks, it induces substantially weaker attacks and consistently results in higher victim win rates than BHEA. This performance gap reflects optimization difficulties rather than representational limitations.

## D. Comparison with Flat Adversarial Policies

In this appendix, we compare our hierarchical adversarial framework with a flat policy baseline. We focus on two settings: (i) the case where at most one agent can be attacked per timestep ($K = 1$), and (ii) the extension to multiple simultaneous attacks ($K > 1$).

## D.1. Flat Adversarial Policy

We consider a flat adversarial policy that directly parameterizes a joint distribution over attack decisions at each timestep. Under the constraint that at most one agent can be attacked per timestep, the flat policy operates over the following discrete action space:

$$\mathcal{A}_{\text{flat}} = \{noop\} \cup \{(i, a) \mid i \in \{1, \ldots, N\},\ a \in \{1, \ldots, A\}\}, \tag{11}$$

where $N$ denotes the number of agents in the environment and $A$ denotes the size of the per-agent action space. The total number of flat actions is therefore $|\mathcal{A}_{\text{flat}}| = 1 + N \times A$.

At each timestep, the adversary samples a single categorical action from this joint space. If noop is selected, no attack is applied. Otherwise, the adversary attacks agent $i$ by replacing its original action with $a$. Each non-noop action consumes one unit of the attack budget. The flat policy is trained using PPO with a standard categorical likelihood and entropy regularization.

Notably, when restricted to $K = 1$, the flat policy has equivalent expressive power to hierarchical attack policies that factorize the decision into attack timing, victim selection, and action replacement. Both parameterizations induce distributions over the same set of $(i, a)$ attack outcomes.

Despite their equivalent expressivity for $K = 1$, we observe a consistent performance gap between flat and hierarchical adversarial policies in practice. Figure 4 reports the victim win rate under attack on representative SMAC maps, evaluated with identical budget schedules, victim policies, and training budgets. Unless otherwise specified, all hyperparameters follow those in Table 5; the only exception is the 3s_vs_3z map, where we set $B = 4$.

Across all tested scenarios, the flat adversary induces substantially weaker attacks than the hierarchical approach, resulting in consistently higher victim win rates. This performance degradation is particularly pronounced on coordination-intensive maps with sparse attack budgets. We attribute this gap to optimization challenges rather than representational limitations. In the flat policy, victim selection and action replacement are tightly coupled into a single categorical decision, leading to higher gradient variance and less effective credit assignment during training. In contrast, the hierarchical design introduces inductive structure that decouples these sub-decisions, resulting in more effective optimization and stronger attacks.

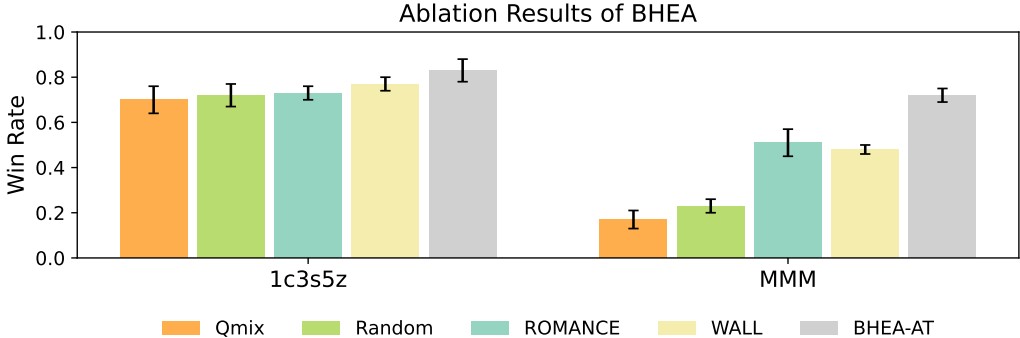

*Figure 5.* Robustness under degraded initial conditions (80% initial health).

## D.2. Scalability to Multiple Attacks per Timestep

When extending the attack setting to allow multiple agents to be attacked simultaneously at a single timestep ($K > 1$), flat policies become impractical. In this regime, the adversary must select a subset of agents along with corresponding replacement actions. A flat parameterization would require an action space of size

$$|\mathcal{A}_{\text{flat}}^{(K)}| = \sum_{k=0}^{K} \binom{N}{k} A^k, \tag{12}$$

which grows combinatorially with both the number of agents and the maximum number of attacks. Even for moderate values of $N$ and $K$, this results in an intractably large action space and renders PPO-style optimization infeasible.

*Table 6.* Victim win rate (%) under full-access BHEA and restricted-access BHEA. In the restricted-access setting, the attacker can only observe and attack within a randomly selected subset of four agents.

| Method | 2s3z | 8m | 1c3s5z | MMM |
|---|---|---|---|---|
| BHEA (full access) | $10.2 \pm 3.5$ | $11.3 \pm 2.5$ | $22.1 \pm 5.5$ | $8.1 \pm 3.2$ |
| BHEA (restricted access) | $15.2 \pm 3.8$ | $37.2 \pm 5.5$ | $42.2 \pm 7.7$ | $26.2 \pm 4.1$ |

By contrast, our hierarchical formulation naturally supports $K > 1$ through sequential victim selection and action replacement, while maintaining a linear dependence on $N$ and $A$. This scalability advantage enables our method to model richer adversarial behaviors under limited attack budgets, which is not achievable with flat joint policies.

## E. Restricted-Access Attack Evaluation

Our main experiments assume a strong execution-time attacker with team-level observability. This threat model is useful for stress-testing cooperative MARL policies, but it also gives the attacker stronger access than purely local-information or compromised-agent-only attack settings. In particular, under CTDE, the victim policy still executes in a decentralized manner using only local observations, whereas the attacker considered in our main setting is an external runtime adversary specified by the threat model. Thus, the stronger assumption lies in the attacker's access, rather than in the victim's decentralized execution protocol.

To further examine the impact of attacker access, we evaluate a restricted-access variant of BHEA on 2s3z, 8m, MMM, and 1c3s5z. In this variant, the attacker is only allowed to observe and attack within a randomly selected subset of four agents. This setting is strictly weaker than full-access BHEA because both the attacker's observability and its action-replacement scope are reduced. The results are reported in Table 6. A lower victim win rate indicates a stronger attack.

The restricted-access attacker is consistently weaker than the full-access attacker, as reflected by the higher victim win rates. This confirms that team-level observability improves attack strength. Nevertheless, BHEA still produces non-trivial attacks under substantially more restrictive access, suggesting that its effectiveness does not fundamentally rely on full team-level observability.

## F. Additional Computational Cost Analysis

In the main paper, we report the wall-clock training time of different adversarial-training methods. However, wall-clock time alone may not fully characterize computational efficiency, as it can be affected by implementation details, hardware utilization, and system-level overhead. Therefore, we further report two additional computational cost metrics: average training FLOPs per update and peak GPU memory usage.

All methods are evaluated under the same hardware setup and the same total training episode budget as in the main paper. For BHEA-AT, 30% of the total episodes are allocated to attacker training over 10 alternating rounds. For ROMANCE and WALL, their training procedures are not based on the same attacker-victim alternating decomposition, so an exactly analogous attacker-training split is not directly definable. We therefore use matched total episode budget as the fairness criterion across methods.

For each method on each SMAC map, we profile full training updates within the actual training pipeline and report the average FLOPs per training update. For victim-training methods, this corresponds to one complete learner update. For methods with additional attacker or population-maintenance components, such as WALL and BHEA-AT, the profiled update also includes the corresponding extra training logic. FLOPs are measured using the PyTorch profiler during training. Since the profiler reports operator-level estimated FLOPs, these values should be interpreted as profiler-estimated training FLOPs rather than exact hardware-level FLOPs. Peak GPU memory is measured using `torch.cuda.max_memory_allocated()`, which records the maximum tensor memory allocated by PyTorch during the profiled training update. The same profiling procedure is applied to all methods for a fair comparison.

As shown in Tables 7 and 8, BHEA-AT is substantially more efficient than WALL in both training FLOPs and peak GPU memory usage across all six SMAC maps. Compared with ROMANCE, BHEA-AT also uses much less peak GPU memory on all maps, and requires fewer training FLOPs in most cases.

*Table 7.* Average FLOPs per training update on six SMAC maps. Values are reported in $10^9$ FLOPs and are profiler-estimated within the actual training pipeline.

| Method | 3m | 8m | 2s3z | 3s_vs_3z | 1c3s5z | MMM |
|---|---|---|---|---|---|---|
| Vanilla QMIX | 0.740 | 2.114 | 2.135 | 1.687 | 12.760 | 7.112 |
| Random | 0.740 | 2.114 | 2.135 | 1.687 | 12.760 | 7.112 |
| ROMANCE | 14.210 | 75.680 | 63.270 | 51.830 | 702.600 | 810.100 |
| WALL | 104.100 | 138.600 | 117.100 | 106.000 | 147.900 | 160.300 |
| BHEA-AT | 9.191 | 28.540 | 18.380 | 9.493 | 43.320 | 47.670 |

*Table 8.* Peak GPU memory usage (GB) on six SMAC maps.

| Method | 3m | 8m | 2s3z | 3s_vs_3z | 1c3s5z | MMM |
|---|---|---|---|---|---|---|
| Vanilla QMIX | 0.033 | 0.076 | 0.051 | 0.032 | 0.089 | 0.092 |
| Random | 0.033 | 0.076 | 0.051 | 0.032 | 0.089 | 0.092 |
| ROMANCE | 0.062 | 0.175 | 0.118 | 0.063 | 0.190 | 0.191 |
| WALL | 0.418 | 0.955 | 0.553 | 0.484 | 1.025 | 1.365 |
| BHEA-AT | 0.037 | 0.080 | 0.055 | 0.039 | 0.094 | 0.098 |

## G. Robustness under Degraded Conditions

We evaluate robustness under degraded initial conditions by initializing all agents with $80\%$ of their default health points at the beginning of each episode, while keeping all other environment settings unchanged. This setting increases the sensitivity to early mistakes and coordination failures, providing a more challenging test for robust policy execution.

Figure 5 reports the resulting test win rates on 1c3s5z and MMM. Under this degraded initialization, all baseline methods exhibit noticeable performance drops compared to standard settings, particularly on the coordination-intensive MMM scenario. In contrast, BHEA-AT consistently achieves higher win rates across both maps, indicating improved robustness and a stronger ability to maintain coordination when the team starts from a disadvantaged state.

## H. Additional Results on MPE and Policy-Gradient Victims

In this appendix, we provide additional experimental results to evaluate the generality of BHEA and BHEA-AT beyond the SMAC benchmark. Specifically, we consider Multi-Agent Particle Environments (MPE), which differ substantially from SMAC in terms of dynamics, observation structure, and coordination patterns. All victim policies in this section are trained using QMIX, and the goal of these experiments is to verify that the effectiveness of our approach does not rely on SMAC-specific environment characteristics.

### H.1. Environments and Experimental Setup

We conduct experiments on two representative MPE tasks: simple_spread and simple_tag.

**Simple Spread.** The simple_spread task is a fully cooperative environment that requires agents to coordinate their movements to cover multiple landmarks while avoiding collisions. We evaluate three different team sizes with the number of agents set to $n \in \{3, 4, 5\}$. This setting allows us to assess how the effectiveness of BHEA scales with increasing coordination complexity. Each episode is truncated at 25 timesteps. The team reward is defined as the negative sum of Euclidean distances from each landmark to its nearest agent, with an additional collision penalty applied when agents collide, following the standard simple_spread environment specification:

$$r_t^{\text{spread}} = -\sum_{\ell \in \mathcal{L}} \min_{i \in \mathcal{A}} \|\mathbf{p}_{i,t} - \mathbf{p}_{\ell,t}\|_2 \; - \sum_{\substack{i,j \in \mathcal{A} \\ i < j}} \mathbb{I}[collision(i,j)], \tag{13}$$

where $\mathcal{A}$ denotes the set of agents, $\mathcal{L}$ the set of landmarks, $\mathbf{p}_{i,t}$ and $\mathbf{p}_{\ell,t}$ their positions at timestep $t$.

**Simple Tag.** We additionally evaluate our method on the simple_tag environment, which features different interaction dynamics from fully cooperative tasks. In this setting, predator agents aim to capture prey agents, while the prey follow

random policies and are not modeled as learning agents. We treat the predator team as the victim and train its policy, while the prey team follows a random policy. We evaluate two configurations: *1 prey vs. 3 predators* and *2 preys vs. 6 predators*. Each episode lasts for 100 timesteps. The predator team receives a positive reward when a prey is captured, along with distance-based shaping rewards that encourage predators to approach the prey, as defined in the standard simple_tag environment:

$$r_t^{\text{tag}} = 10 \sum_{g \in \mathcal{G}} \sum_{p \in \mathcal{P}} \mathbb{I}[collision(g, p)] \; - \; \sum_{p \in \mathcal{P}} \left\| \mathbf{p}_{p,t} - \mathbf{p}_{g^*(p),t} \right\|_2, \tag{14}$$

where $\mathcal{P}$ and $\mathcal{G}$ denote the sets of predators and prey, $\mathbf{p}_{p,t}$ and $\mathbf{p}_{g,t}$ their positions, and $g^*(p)$ is the prey closest to predator $p$ at timestep $t$.

*Table 9.* Attack capacity $K$ and episode-level attack budget $B$ for MPE environments. For simple_tag, the notation $avb$ denotes $a$ prey agents and $b$ predator agents; predators are treated as the victim team, while prey follow random policies.

| Scenario | Attack Capacity $K$ | Attack Budget $B$ |
|---|---|---|
| simple_spread ($n = 3$) | 2 | 5 |
| simple_spread ($n = 4$) | 2 | 5 |
| simple_spread ($n = 5$) | 3 | 5 |
| simple_tag (1v3) | 2 | 25 |
| simple_tag (2v6) | 4 | 25 |

Across all environments, we compare BHEA and BHEA-AT against representative baseline attacks and adversarial training methods under matched attack budgets, with the specific attack capacity $K$ and episode-level budget $B$ given in Table 9. Other hyperparameters follow the same settings as in the SMAC experiments and are kept consistent across methods. The total training budget is 1,000,000 episodes for simple_spread and 200,000 episodes for simple_tag. For adversarial training with BHEA-AT, training proceeds in an alternating manner for 10 iterations: in each iteration, the adversary is trained for 50,000 (10,000) episodes on simple_spread (simple_tag), followed by training the victim policy for the same number of episodes.

Under a limited episode-level budget $B$, perturbing the action of a single agent often has only a marginal impact on the overall team reward in MPE environments. To induce meaningful performance degradation, we therefore allow the adversary to hijack multiple agents per timestep by setting $K > 1$. Since ROMANCE and EGA do not support multi-agent action perturbations within a single timestep, these methods are not included in this experiment.

*Table 10.* Performance of policy-gradient victim algorithms under natural execution and BHEA attacks on MPE environments defined in Section H. Results are reported as mean $\pm$ estimated standard deviation.

| Victim | Environment | Natural | BHEA |
|---|---|---|---|
| MAPPO | simple_spread ($n = 3$) | $-14.8 \pm 0.6$ | $-19.1 \pm 1.1$ |
| MAPPO | simple_spread ($n = 4$) | $-21.2 \pm 0.5$ | $-24.3 \pm 1.2$ |
| MAPPO | simple_spread ($n = 5$) | $-24.9 \pm 0.5$ | $-29.6 \pm 1.4$ |
| MAPPO | simple_tag (1v3) | $62.1 \pm 1.9$ | $45.0 \pm 1.6$ |
| MAPPO | simple_tag (2v6) | $76.4 \pm 2.2$ | $58.2 \pm 1.0$ |
| MADDPG | simple_spread ($n = 3$) | $-15.5 \pm 0.6$ | $-19.4 \pm 1.2$ |
| MADDPG | simple_spread ($n = 4$) | $-22.1 \pm 0.4$ | $-25.2 \pm 1.3$ |
| MADDPG | simple_spread ($n = 5$) | $-26.0 \pm 0.3$ | $-31.1 \pm 1.5$ |
| MADDPG | simple_tag (1v3) | $60.3 \pm 2.0$ | $42.6 \pm 2.8$ |
| MADDPG | simple_tag (2v6) | $74.8 \pm 2.3$ | $55.7 \pm 3.2$ |

## H.2. Generalization of BHEA and BHEA-AT on MPE

The results on MPE benchmarks closely mirror the trends observed on SMAC, as summarized in Table 11. Across all evaluated environments and agent configurations, BHEA consistently induces substantially stronger performance degradation than random and Wolfpack attacks, demonstrating the effectiveness of budget-aware hierarchical coordination

*Table 11.* Performance comparison on MPE environments under different adversarial attack settings. Results are reported as mean $\pm$ estimated standard deviation. For simple_tag, the notation $avb$ denotes $a$ prey agents and $b$ predator agents; predators are treated as the victim team, while prey follow random policies.

| Attack Setting | Environment | simple_spread $(n=3)$ | simple_spread $(n=4)$ | simple_spread $(n=5)$ | simple_tag (1v3) | simple_tag (2v6) |
|---|---|---|---|---|---|---|
| Natural | Vanilla QMIX | $-15.2 \pm 0.3$ | $-20.1 \pm 0.4$ | $-23.7 \pm 0.5$ | $58.2 \pm 2.2$ | $74.1 \pm 2.5$ |
| | RANDOM | $-15.1 \pm 0.6$ | $-20.2 \pm 0.4$ | $-23.5 \pm 0.9$ | $58.4 \pm 2.3$ | $74.3 \pm 2.4$ |
| | WALL | $-15.1 \pm 0.7$ | $-19.5 \pm 0.2$ | $-23.1 \pm 0.4$ | $58.6 \pm 2.1$ | $74.6 \pm 3.6$ |
| | **BHEA-AT** | $\mathbf{-14.9 \pm 0.2}$ | $\mathbf{-19.4 \pm 0.3}$ | $\mathbf{-22.9 \pm 0.4}$ | $\mathbf{58.7 \pm 2.0}$ | $\mathbf{75.2 \pm 3.2}$ |
| Random Attack | Vanilla QMIX | $-15.7 \pm 0.5$ | $-20.7 \pm 0.6$ | $-24.5 \pm 0.7$ | $52.4 \pm 2.8$ | $67.2 \pm 2.1$ |
| | RANDOM | $-15.3 \pm 0.4$ | $-20.2 \pm 0.5$ | $-24.0 \pm 0.6$ | $54.4 \pm 2.6$ | $69.5 \pm 2.9$ |
| | WALL | $-15.2 \pm 0.4$ | $-19.9 \pm 0.5$ | $-23.5 \pm 0.6$ | $55.2 \pm 2.5$ | $71.2 \pm 2.8$ |
| | **BHEA-AT** | $\mathbf{-15.0 \pm 0.3}$ | $\mathbf{-19.5 \pm 0.4}$ | $\mathbf{-23.1 \pm 0.5}$ | $\mathbf{56.6 \pm 2.3}$ | $\mathbf{72.7 \pm 3.6}$ |
| Wolfpack Attack | Vanilla QMIX | $-16.8 \pm 0.8$ | $-21.3 \pm 0.9$ | $-25.8 \pm 1.1$ | $47.2 \pm 2.5$ | $59.6 \pm 2.8$ |
| | RANDOM | $-15.9 \pm 0.7$ | $-20.9 \pm 0.8$ | $-24.9 \pm 0.9$ | $49.8 \pm 2.2$ | $61.4 \pm 2.4$ |
| | WALL | $-15.6 \pm 0.6$ | $-20.4 \pm 0.7$ | $-24.2 \pm 0.8$ | $51.1 \pm 2.0$ | $64.7 \pm 2.1$ |
| | **BHEA-AT** | $\mathbf{-15.2 \pm 0.5}$ | $\mathbf{-20.1 \pm 0.6}$ | $\mathbf{-23.7 \pm 0.7}$ | $\mathbf{55.6 \pm 1.5}$ | $\mathbf{66.8 \pm 1.8}$ |
| BHEA | Vanilla QMIX | $-19.2 \pm 1.8$ | $-23.2 \pm 2.1$ | $-28.4 \pm 2.5$ | $40.6 \pm 3.4$ | $55.7 \pm 3.8$ |
| | RANDOM | $-18.3 \pm 1.5$ | $-22.4 \pm 1.8$ | $-27.6 \pm 2.2$ | $42.9 \pm 3.1$ | $58.1 \pm 3.5$ |
| | WALL | $-18.2 \pm 1.2$ | $-21.7 \pm 1.5$ | $-25.7 \pm 1.9$ | $49.2 \pm 2.5$ | $63.2 \pm 2.8$ |
| | **BHEA-AT** | $\mathbf{-16.7 \pm 0.8}$ | $\mathbf{-20.1 \pm 1.0}$ | $\mathbf{-23.6 \pm 1.2}$ | $\mathbf{54.2 \pm 1.8}$ | $\mathbf{69.8 \pm 2.0}$ |

in disrupting cooperative policies. Importantly, this advantage persists as the number of agents increases, showing that the attack remains effective under growing coordination complexity.

Conversely, policies trained with BHEA-AT achieve the best performance under all attack settings, significantly outperforming vanilla QMIX and baseline adversarial training methods. This indicates that alternating training against a budget-aware hierarchical adversary leads to substantially improved robustness against limited-step action hijacking.

Overall, these results demonstrate that the advantages of BHEA and BHEA-AT are not confined to SMAC, but generalize well across different MPE tasks, team sizes, and interaction dynamics, supporting their use as an effective and environment-agnostic framework for robustness evaluation and training in cooperative multi-agent reinforcement learning.

### H.3. Additional Results on Policy-Gradient Victims

To further evaluate whether the effectiveness of BHEA depends on value-decomposition-based learning, we additionally test BHEA against policy-gradient multi-agent reinforcement learning algorithms. Specifically, we consider MAPPO and MADDPG as victim policies, both of which rely on centralized critics and decentralized actors, but differ fundamentally from QMIX in their optimization objectives.

We conduct experiments on multiple MPE environments with diverse coordination and interaction patterns, including simple_spread and simple_tag, under different agent configurations. Victim policies follow standard MAPPO and MADDPG implementations without any architectural modification. The adversary uses the same budget-aware hierarchical attack as in previous experiments, with identical attack capacity $K$, episode-level budget $B$, and action hijacking mechanism.

In this set of experiments, we focus exclusively on comparing natural execution and execution under BHEA, without including baseline attacks. Baseline attack methods have already been extensively evaluated with QMIX across SMAC and MPE environments. Excluding them here allows us to isolate whether the effectiveness of BHEA depends on the underlying learning paradigm, rather than re-evaluating baseline strategies.

The results are summarized in Table 10. Across all evaluated environments and victim algorithms, BHEA induces substantial performance degradation compared to natural execution. This observation holds consistently across different coordination structures and agent counts, indicating that the effectiveness of budget-aware hierarchical attacks does not rely on value decomposition and generalizes naturally to policy-gradient-based multi-agent reinforcement learning methods.

