# OpenReview forum: "Budget-Efficient Attacks and Robustness Training for Cooperative MARL"
_ICML.cc/2026/Conference — ICML 2026 regular_

### Official Review · Reviewer_5Ktv · 2026-03-02

**Soundness:** 3
**Presentation:** 3
**Significance:** 2
**Originality:** 2
**Overall Recommendation:** 4
**Confidence:** 4

**Summary:**

This paper proposed a novel budget-efficient adversarial attack for Cooperative MARL(BHEA). The proposed method decomposed the attack process into two parts: which agents to attack and what adversarial actions to choose. The authors use a hierarchical RL framework and PPO to train the attacker. The paper also shows adversarial training can be applied on top of BHEA to create a more robust MARL policy. The experiments conducted in the SMAC environment show that the proposed attack and robust adversarial training policy achieve the best performance compared to various baselines.

**Compliance With Llm Reviewing Policy:**

Affirmed.

**Final Justification:**

The rebuttal has addressed my concerns.

**Key Questions For Authors:**

## Key Questions

1. **Threat model realism/access assumptions.**

The attacker appears to observe joint agent information (the method encodes each agent’s observation together with a mean-aggregated global context), which may require broad system access during execution.

Could the authors clarify what level of access is assumed in practice (e.g., access to all agents’ observations vs. only local observations, and what level of control access is needed over agents). Relatedly, how would performance change in a gray-box/black-box setting where the attacker has partial observability or limited access?

2. **Evaluation scope/generality.**

The SMAC evaluation is thorough, but the empirical coverage would be stronger with additional domains that better reflect real-world cooperative systems (e.g., multi-agent autonomous driving, embodied/image-based environments, or other partially observable continuous-control settings). The paper includes additional results on MPE in the appendix (e.g., *simple spread* and *simple tag*), but it would help to see broader and more realistic benchmarks to support generalization claims.

3. **Robustness against diffusion-based defenses.**

Recent robust-RL defenses in the single-agent setting leverage diffusion models[1], and diffusion policies have also been explored for MARL[2]. It would strengthen the paper to discuss whether BHEA-style action hijacking can bypass diffusion-based robustness mechanisms (e.g., denoising/trajectory priors, diffusion-based policy smoothing), and—if possible—to include a preliminary evaluation against such defenses or a reasoned argument for expected transferability.

[1] Yang & Xu, DMBP: Diffusion Model-Based Predictor for Robust Offline Reinforcement Learning against State Observation Perturbations. ICLR 2024.

[2] Zhu et al., MADiff: Offline Multi-agent Learning with Diffusion Models. NeurIPS 2024.

**Limitations:**

The paper does not include a dedicated limitations section. I suggest adding one to discuss the strong attacker access assumptions and the limited diversity of evaluation environments (primarily SMAC), as both factors may affect the realism and generalizability of the results.

**Strengths And Weaknesses:**

## Strengths

**Soundness.**
The proposed budget-efficient hierarchical framework is well-motivated and technically reasonable. Decomposing attack timing/victim selection from action replacement reduces optimization complexity. The experimental results and ablations support the effectiveness of this design.

**Presentation.**
The paper is clearly written and easy to follow. The methodology and empirical results are well organized.

**Significance.**
The work addresses robustness in cooperative MARL, an increasingly important area as multi-agent systems become more prevalent in real-world applications.

**Originality.**
The hierarchical RL problems formulation appears novel within cooperative MARL robustness.

## Weaknesses

**Significance.**
The evaluation is primarily conducted on SMAC and MPE, which are game-based benchmarks. Additional experiments in more diverse or realistic environments would strengthen the practical impact.

**Originality.**
The two-stage decomposition (deciding when/how many to perturb and then how to perturb) has been explored in other domains. While its adaptation to MARL robustness is meaningful, the high-level idea is not entirely new. Such as [1] in the voting field, which also decomposes the problem into how many actions to change and how to change the actions in an optimal way.

[1] Sun et.al., Pandering in a (flexible) representative democracy. UAI 2023.

---

> ### Author Rebuttal · Authors · 2026-03-27
>
> ### Response to Q1
> > Thank you for this important comment. We will clarify the threat model more explicitly in the revision. Our attacker is **black-box with respect to the victim policy**: it does not assume access to victim parameters, gradients, architecture, or training procedure, and only uses execution-time inputs/outputs. At the same time, our current setting assumes relatively strong **runtime access**: the attacker can observe the victim team’s current observations and replace the executed actions of selected agents under per-step and episode-level budget constraints. Thus, the setting is black-box in **model knowledge**, but not minimal-access in **execution authority/observability**.
> To study a weaker-access setting, we additionally evaluated a **restricted-access variant** on 2s3z, 8m, MMM, and 1c3s5z, where the attacker can only observe and attack within a randomly selected subset of 4 agents; the subset is resampled across trials and results are averaged. This is a strictly weaker threat model, since both observability and control scope are restricted. Even under this restricted-access setting, the attacker remains effective. As shown in Table A, restricting both observability and attack scope to a random subset of 4 agents consistently weakens the attack, but does not eliminate it. This shows that full-team access strengthens attack performance, but BHEA still learns non-trivial attacks under substantially more restrictive access.
>
> >**Table A. Victim win rate (%) under full-access BHEA and restricted-access BHEA (subset of 4 agents).**
> | Method | 2s3z | 8m | 1c3s5z | MMM |
> |---|---:|---:|---:|---:|
> | BHEA (full access) | 10.2±3.5 | 11.3±2.5 | 22.1±5.5 | 8.1±3.2 |
> | BHEA (restricted access) | 15.2±3.8 | 37.2±5.5 | 42.2±7.7 | 26.2±4.1 |
>
> ### Response to Weakness 2
> > Thank you for this helpful comment. We agree that hierarchical or staged decision structures have appeared in prior work in other domains. Our contribution is not the decomposition idea itself, but its specific formulation for budget-constrained execution-time attacks in cooperative MARL. In our setting, the attacker must jointly decide when/how many agents to perturb under a limited budget and how to modify their actions to maximally disrupt team coordination. This leads to a distinct optimization problem from prior work in other domains. What is non-trivial here is handling sequential budget allocation, partial observability, and coordinated victim/action coupling in real time.
> The cited work studies pandering in voting, whereas our paper focuses on real-time adversarial action perturbation in MARL. Although both involve a high-level allocation decision followed by a low-level action decision, the task setting, threat model, constraints, and learning problem are substantially different.
> We will clarify this in the revision by emphasizing that our novelty lies in adapting and instantiating this hierarchy for MARL robustness, where it improves learnability and empirical attack effectiveness under combinatorial budget constraints.
>
> ### Response to Weakness 1 and Q2
> >Thank you for the suggestion. We agree that additional evaluation in more realistic domains could be valuable. However, for the problem studied in this paper, we believe SMAC and MPE already provide two representative and complementary cooperative MARL benchmarks. SMAC is a standard testbed with strong partial observability, challenging coordination, and large combinatorial joint action spaces, while MPE offers a different cooperative multi-agent setting with distinct dynamics and interaction patterns. Evaluating on both therefore allows us to test the method across two widely used benchmark families rather than a single environment class.
> We will clarify in the revision that our empirical claims are restricted to these established cooperative MARL benchmarks, and we will avoid overstating broader real-world generalization. Exploring more realistic domains is an important direction for future work.
>
> ### Response to Q3
> >Thank you for this insightful suggestion. We will add a discussion of diffusion-based robust RL in the revision. However, the threat model is different from ours. In particular, DMBP focuses on robustness to state observation perturbations in single-agent offline RL, while our method studies execution-time action hijacking in cooperative MARL. More importantly, BHEA attacks the executed action after policy inference, whereas diffusion-based methods such as DMBP are designed to improve robustness to perturbed state observations before action generation. Therefore, they do not directly address the threat model considered in our paper. Similarly, MADiff studies diffusion-based multi-agent policy learning rather than defenses against adversarial action replacement. Whether diffusion-based MARL policies provide additional robustness to this type of attack would require a dedicated study and is beyond the scope of the current paper.

---

> > ### Author Rebuttal · Reviewer_5Ktv · 2026-04-01
> >
> > Thanks for the detailed response. All my concerns have been resolved.

---

### Official Review · Reviewer_ZDuH · 2026-03-11

**Soundness:** 3
**Presentation:** 2
**Significance:** 3
**Originality:** 3
**Overall Recommendation:** 3
**Confidence:** 4

**Summary:**

This manuscript proposes a novel robustness training scheme for CMARL, which enhances resistance against limited-step action hijacking while simultaneously reducing training overhead. Extensive experiments conducted on the SMAC and MPE benchmarks validate the effectiveness and efficiency of the proposed scheme.

**Compliance With Llm Reviewing Policy:**

Affirmed.

**Final Justification:**

Based on the authors' response, I decide to raise the Significance score from 2 to 3. However, as my primary concerns remain unresolved (please refer to my Official Review and Rebuttal Acknowledgement), I will maintain my Overall Recommendation.

**Key Questions For Authors:**

Q1. The threat model discussed in this manuscript appears somewhat impractical. Specifically, it assumes that an attacker can arbitrarily manipulate the actions of victim agents and has full access to their joint observations. Could the authors provide concrete, real-world scenarios that align with these assumptions?
Furthermore, does discussing a limited attack budget remain meaningful under such a permissive setting? If the attacker already possesses the capability to arbitrarily manipulate victim actions and access joint observations, the constraints imposed by a 'budget' seem contradictory or redundant. Could the authors clarify the underlying rationale for combining these assumptions?

Q2. Is the proposed robustness training (Algorithm 1) fundamentally distinct from existing self-play frameworks [1][2][3]? I encourage the authors to explicitly distinguish their algorithmic design from these classic paradigms to clarify the originality of their contribution.
Furthermore, under identical attack settings, does the proposed robustness training incur lower computational overhead compared to the baselines? If so, what are the underlying reasons for this efficiency gain?
Does the design of Eq. (9) effectively guarantee that training is mildly biased toward stronger adversaries? Is there a risk that the adversaries might collapse during the training process? I suggest the authors provide an analysis or empirical evidence regarding the training stability of the adversary pool.

&nbsp;  [1] Bansal T., et al. "Emergent complexity via multi-agent competition." ICLR, 2018.

&nbsp;  [2] Berner C., et al. "Dota 2 with large scale deep reinforcement learning." arXiv, 2019.

&nbsp;  [3] Jaderberg M., et al. "Human-level performance in 3D multiplayer games with population-based reinforcement learning." Science, 2019.

Q3. The term 'adversarial policies' has established definitions in specific contexts [4][5][6], and its usage here may lead to unnecessary confusion. I suggest the authors replace it with more precise terminology that better reflects their specific setting, such as 'hijacking policies'.

&nbsp;  [4] Gleave, A., et al. "Emergent complexity via multi-agent competition." ICLR, 2018.

&nbsp;  [5] Guo W., et al. "PATROL: Provable Defense against Adversarial Policy in Two-player Games." USENIX Security, 2023.

&nbsp;  [6] Ma O., et al. "SUB-PLAY: Adversarial Policies against Partially Observed Multi-Agent Reinforcement Learning Systems." CCS, 2024.

Q4. I am curious about the relationship between limited-step and non-limited-step action hijacking. Does an agent trained against limited-step action hijacking exhibit any degree of robustness when facing non-limited-step action hijacking? What would be the outcome if the robustness training simultaneously accounted for both limited-step and non-limited-step action hijacking?

**Limitations:**

yes

**Strengths And Weaknesses:**

Strengths:

S1. The proposed scheme sounds make sense within the scope of the defined threat model.

S2. Extensive experiments across SMAC and MPE effectively demonstrate the performance gains and the overall effectiveness of the proposed method.

Weaknesses:

W1. Impractical Threat Model.

W2. The proposed robustness training exhibits no significant departure from traditional self-play.

---

> ### Author Rebuttal · Authors · 2026-03-28
>
> ### Response to Q1 and W1
> >Thank you for the question. We would like to clarify that the threat model in our paper is not **introduced ad hoc, but follows the execution-time attack setting commonly adopted in prior MARL attack literature**, such as Wolfpack[1] and ROMANCE[2]. Therefore, this is not a deliberately relaxed assumption made to amplify attack performance, but a standard starting point consistent with existing work.
> From a practical perspective, this setting can correspond to scenarios in which the control pipeline, middleware, or execution interface of systems such as drone swarms, autonomous vehicle platoons, or warehouse robots is compromised, allowing an attacker to temporarily manipulate the executed actions of a subset of agents during deployment.
> Meanwhile, attack capability and budget constraints are not contradictory. The former defines what the attacker can do in a single intervention, while the latter defines how often it can intervene, how many agents it can affect, and for how long. **In practice, even an attacker with strong execution-side access is still constrained by stealth, bandwidth, energy cost, and exposure risk.** Therefore, the budget remains both meaningful and necessary.
> ### Response to Q2 and W2
> >Thank you for the comment. We would like to clarify that our method is adversarial training, not self-play. These are different concepts from different research contexts, and adversarial training is already a widely used paradigm in adversarial attack and robustness research[1-4]. Therefore, we do not claim the overall training framework as the main novelty of this paper.
>
> >Our main contribution is to develop a more effective and computationally efficient attacker training method for the budget-constrained execution-time attack setting studied here. As shown in Table 2, our method has lower training time cost. In addition, in our response to Reviewer rDca, we further measured training FLOPs and peak GPU memory, and both are also lower than the baselines under the same attack setting.
>
> >For Eq. (9), its purpose is to assign larger sampling weights to newly introduced attackers that still need training, while keeping smaller but non-zero weights for previous attackers to prevent the victim from forgetting earlier attack patterns. This is because newly introduced attackers are usually the hardest ones for the current victim, whereas earlier attackers have already been seen during adversarial training and the victim has therefore acquired partial robustness against them.
>
> >Finally, regarding adversary collapse, we agree this is an important concern. Empirically, we did not observe collapse in practice: the adversary pool maintained non-trivial attack effectiveness throughout training and led to stable robustness gains.
> ### Response to Q3
> >Thank you for the suggestion. We agree that “adversarial policies” may be confusing because this term already has specific meanings in prior work. Our setting is more precisely execution-time action hijacking under a budget constraint. To avoid ambiguity, we will revise the paper and use more precise terms such as “hijacking policies” or “hijacking attackers.”
> ### Response to Q4
> >Thank you for this interesting question. We view limited-step and non-limited-step hijacking as related threat models with different attack budgets, where non-limited-step hijacking is strictly stronger. A victim trained against limited-step hijacking may therefore show some partial robustness under non-limited-step hijacking, but this robustness is expected to be limited because the test-time attacker is much stronger than the one used in training.
> More importantly, from the defense perspective, unlimited hijacking can make robust training itself lose meaning. If the attacker can hijack all agents with no limit on the number of attacked steps, the victim policy is essentially overridden throughout execution, leaving little room for the victim to adapt or improve under attack. Even if the number of attacked agents is limited, removing the step budget can still make the attacker too strong for effective adversarial training. Therefore, to use hijacking policies as training adversaries, one must control attack strength. In this sense, limiting the number of attack steps is important not only for modeling realistic attackers, but also for making robustness training feasible and meaningful.
>
> [1]Lee, S., et al. Wolfpack adversarial attack for robust multi-agent reinforcement learning. ICML, 2025.
>
> [2]Yuan, L., et al. Robust multi-agent coordination via evolutionary generation of auxiliary adversarial attackers. AAAI, 2023.
>
> [3]Zhang, H., et al. Robust Reinforcement Learning on State Observations with Learned Optimal Adversary. ICLR, 2021.
>
> [4]Reddi, A., et al. Robust Adversarial Reinforcement Learning via Bounded Rationality Curricula. ICLR, 2024.

---

> > ### Author Rebuttal · Reviewer_ZDuH · 2026-04-02
> >
> > Thanks to the authors for the detailed rebuttal, which has addressed part of my concerns. However, I still have the following three reservations:
> >
> > - Assuming that, in a MARL system, a specific agent may be compromised under certain conditions appears reasonable. However, this work considers a significantly stronger setting where the adversary can arbitrarily manipulate the action outputs of any agent at any time. I remain unconvinced about the practical realism of such an assumption, even though similar threat models have been discussed in prior work.
> >
> > - If I understand correctly, the goal of action hijacking is to degrade the performance of the target MARL system as much as possible. Under this objective, I do not see why stealth or exposure risk would be relevant factors, as they typically arise in settings such as backdoor attacks. Therefore, I find it difficult to agree with the statement: "In practice, even an attacker with strong execution-side access is still constrained by stealth, bandwidth, energy cost, and exposure risk." unless the authors can explicitly justify this claim. For example, by clearly defining what “stealth” means in this context and providing a way to measure it.
> >
> > - I am not questioning whether Algorithm 1 qualifies as adversarial training. Rather, I would appreciate a clearer explanation of how its implementation differs from standard self-play, as the two appear highly similar in form. Based on my current understanding, Algorithm 1 does not seem to be fundamentally different from existing frameworks. Clarifying this distinction would help readers better assess the paper’s contribution.
> >
> > Given the above concerns, I will maintain my current rating.

---

> > > ### Author Response · Authors · 2026-04-02
> > >
> > > Thank you for the follow-up. We appreciate the clarification of your remaining concerns. We will revise the paper to make these points more precise.
> > > ### Response to Q1
> > > >We would like to explain the rationale of our threat model from both the defense and attack perspectives.
> > >
> > > >From the defense perspective, studying adversarial attacks is not only about breaking the system, but also about exposing weaknesses of already-trained policies and improving them through adversarial training. Under this use case, our assumption is meaningful: during robustness training, we do have full control over the victim team and full access to its information, so a strong execution-time hijacking attacker serves as a valid stress test. More importantly, the robustness learned in this way is not limited to the specific hijacking attacker used in training. In Figure 1, BHEA-AT improves performance under unseen attacks across multiple victim learners, and Appendix E further shows that under degraded initial conditions (all agents initialized with 80% health), BHEA-AT still achieves the highest win rates among all baselines. This suggests that the learned robustness has a degree of generalization, rather than being specific only to action hijacking.
> > >
> > > >From the attack perspective, we agree that arbitrary execution-time action override is a strong capability assumption. Our point is not that it universally holds in all deployments, but that it corresponds to an important execution-layer threat surface: for example, if the cloud server, middleware, or low-level execution interface of a multi-agent system is compromised, an attacker may indeed be able to override issued actions during deployment. Under such a threat surface, the key practical constraint is often not whether the attacker can ever intervene, but how frequently it can do so without being exposed. If the attacker intervenes too frequently, the resulting abnormal behavior is more likely to be detected. In this sense, limiting the intervention frequency is meaningful, and our formulation studies exactly this regime through the episode-level budget B and per-step capacity K.
> > >
> > > >We will revise the paper to make this scope clearer.
> > >
> > > ### Response to Q2
> > > > We would first like to clarify that some form of attack limitation is necessary for this problem to be meaningful. If execution-time hijacking is unrestricted, the setting becomes close to direct actuator takeover: this is already a very strong assumption, the evaluation becomes almost trivial (continuously overriding executed actions will unsurprisingly break the system), and adversarial training leaves little room to actually improve robustness. This is why our paper studies limited-budget action hijacking, rather than unrestricted continuous hijacking.
> > >
> > > > We also agree that our previous wording about “stealth” was too brief. We did not intend to use “stealth” in a backdoor-specific sense. Rather, our intended meaning was the broader RL notion of sparse, well-timed interventions, rather than continuous interference at every step[1-3]. Prior work on stealthy DRL attacks[1-3] similarly studies attacking only a minimal set of critical moments while still causing severe degradation; in our multi-agent setting, this intuition naturally extends from critical timesteps to critical timesteps and critical agents.
> > >
> > > [1]Sun, J., et al. Stealthy and Efficient Adversarial Attacks against Deep Reinforcement Learning. (AAAI 2020)
> > >
> > > [2]Lin, Y., et al. Tactics of Adversarial Attack on Deep Reinforcement Learning Agents. (IJCAI 2017)
> > >
> > > [3]Cheng, Z., et al. StateMask: Explaining Deep Reinforcement Learning through State Mask. (NeurIPS 2023)
> > >
> > > ### Response to Q3
> > > >Thank you for the clarification. We agree that Algorithm 1 is similar in outer form to standard alternating opponent-training or self-play-style procedures, and we will make this clearer in the revision. Our claim is not that the alternating update itself is the main novelty of the paper.
> > >
> > > > The key distinction lies in the adversarial inner loop. In standard self-play, the opponent is typically another policy trained in the original environment action space. In our setting, however, the opponent is an execution-time hijacker operating in a different decision space: it must decide when to intervene, which agent(s) to hijack, and how to replace their actions, under explicit attack constraints given by the episode-level budget B and per-step capacity K. This is why our attacker is parameterized as a hierarchical policy $\pi_{\text{sel}} \times \pi_{\text{act}}$, rather than a standard self-play policy.
> > >
> > > > Therefore, the contribution of the paper is not the generic alternating training template, but the design of a budget-aware hierarchical hijacking adversary and the resulting adversarial training instantiation built on top of it. We will revise the paper to state this positioning more explicitly.

---

### Official Review · Reviewer_fXz8 · 2026-03-15

**Soundness:** 2
**Presentation:** 2
**Significance:** 2
**Originality:** 2
**Overall Recommendation:** 3
**Confidence:** 3

**Summary:**

The paper focuses on the domain of budget-efficient adversarial attacks and robustness training for cooperative MARL under limited-step action hijacking. Concretely, the paper proposes BHEA, a hierarchical attacker that separates attack timing / victim selection from action replacement, and then builds BHEA-AT on top of it.

**Compliance With Llm Reviewing Policy:**

Affirmed.

**Key Questions For Authors:**

See above

**Limitations:**

yes

**Strengths And Weaknesses:**

## Strengths
- The paper studies an interesting and practically relevant problem.
- The hierarchical decomposition is reasonable. Separating when and which agents to attack from how to replace actions is intuitive and well aligned with the limited-budget setting.
- The empirical section is fairly comprehensive, including SMAC, MPE, unseen-attack evaluations, sensitivity analysis, and ablations.
- The efficiency result is also interesting. BHEA-AT appears substantially cheaper than prior baselines such as WALL and ROMANCE, while still achieving strong robustness.

## Weaknesses
However, I feel the paper still lacks clarity on what is the main technical gain.

- The paper itself states that, for $K=1$, the hierarchical and flat adversarial policies are expressively equivalent. Then the gain seems to be mainly optimization and inductive bias rather than representational power. I think this point should be stated much more clearly.
- The explanation for why the hierarchy helps is a little bit high-level. The paper attributes the gap to optimization difficulty in the flat policy, but I still want a deeper discussion of the actual mechanism. Is it mainly better exploration over scarce attack opportunities, lower gradient variance, or better credit assignment?
- I am also not fully convinced by the generalization claim. Evaluating BHEA-AT under BHEA is a strong stress test, but it is also related to the attack family used during training.
- Finally, I think the model introduced by the paper is somewhat a multi-agent extension of the action-robust rl problem (e.g., Action Robust Reinforcement Learning and Applications in Continuous Control, ICML 2019). I think it is worth more discussions on related literature.

---

> ### Author Rebuttal · Authors · 2026-03-27
>
> We thank the reviewer for the careful reading and constructive feedback. We are glad that the reviewer recognizes the practical relevance of the problem, the reasonableness of the hierarchical decomposition, the breadth of the empirical evaluation, and the favorable efficiency of BHEA-AT relative to prior baselines. We address each concern below.
> ### Response to Weakness 1
> >We agree with the reviewer that this point should be stated much more clearly. In particular, for \(K=1\), we do **not** claim that the advantage of BHEA comes from stronger representational power: as stated in the paper, the hierarchical and flat adversarial policies are expressively equivalent in this case. Our main gain is instead a better optimization structure and inductive bias for budget-limited attacks. A flat policy must jointly learn *whether/whom to attack* and *how to replace actions* in a single decision, while BHEA explicitly decomposes these subproblems, which makes learning substantially more effective in practice. We will revise the paper to make this point explicit in both the method description and the discussion of Appendix D, so that the contribution is more accurately framed as an optimization/scalability advantage rather than a representational one. Moreover, when \(K>1\), this structural advantage becomes even more important, since a flat adversary must optimize over a combinatorially growing joint action space over victim subsets and replacement actions.
> ### Response to Weakness 2
> > We agree that our original explanation was too high-level, and we will revise the paper to make the mechanism clearer. Our view is that the advantage of the hierarchy comes from **both a reduced/structured action space and better reward assignment**. In our setting, a successful attack requires two qualitatively different decisions: (i) identifying a rare timestep/agent where spending one unit of budget is worthwhile, and (ii) choosing a damaging replacement action once that victim is fixed. In a flat policy, these two decisions are entangled into one joint categorical choice, which makes the effective action space much larger. As a result, the return is assigned only to the entire joint event (*who/when* + *how*), so the learner cannot tell whether a poor outcome is caused by wrong timing/victim selection or by a poor replacement action. This makes the feedback highly mixed and sample-inefficient under sparse budgets. BHEA resolves this by explicitly decomposing the attack into selector and attacker. This decomposition reduces the action space faced by each policy, while also improving credit assignment through **separate value functions, advantages, and PPO objectives** for the selector and the attacker. The selector is trained only to evaluate budget-aware timing/victim decisions, whereas the attacker is trained only to evaluate conditional replacement actions once the victim is fixed. This allows the return to be assigned at the proper decision level, instead of being entangled within a single joint action. In practice, this improves credit assignment and leads to lower optimization difficulty and gradient variance. We will revise the paper to make this mechanism explicit.
> ### Response to Weakness 3
> > Thank you for this important comment. We would like to clarify that our generalization claim is mainly supported by **Table 1**, rather than by Figure 1 alone. Table 1 evaluates BHEA-AT under multiple attack baselines, including **BHEA, EGA, and Wolfpack**, and shows that BHEA-AT consistently remains strong across these different attacks. In this sense, it already provides a stronger form of unseen-attack evaluation than testing only within the training attack family. By contrast, the role of **Figure 1** is different: it is mainly intended to show the **training-time trend** of robustness improvement under a weaker unseen-attack setting, i.e., attacks generated by the same algorithmic family but not used during adversarial training. We agree that this distinction should be stated more clearly. In the revision, we will clarify that Table 1 is the main evidence for cross-attack robustness, while Figure 1 serves as an auxiliary result showing how robustness against such unseen attacks improves over training.
> ### Response to Weakness 4
> >We agree that our setting is conceptually related to the action-robust literature, and we will strengthen this discussion in the revision. Our problem can indeed be viewed as extending action-level robustness from single-agent RL to cooperative MARL. At the same time, our setting is substantially different: the adversary must allocate a limited budget across time, decide **when** to attack, **which agents** to hijack, and **how** to replace their actions under explicit per-step and episode-level constraints. These challenges are central to our formulation and motivate the hierarchical attacker design. We will revise the related-work section to clarify both this connection and these key differences.

---

> > ### Author Rebuttal · Reviewer_fXz8 · 2026-04-04
> >
> > I thank the reviewer for the response. However, I still feel the justification of hierarchy is a bit conceptual. I think such benefits cannot come for free. This is a bit analogous to single-agent RL v.s. multi-agent RL. Single-agent RL for multi-agent formulation also faces the issues of exponentially large action space. However, for multi-agent RL to solve this, explicit coordination is needed and it is more easily to converge to local optima. In other words, I think it should be better to measure the optimization difficulty more directly.

---

> > > ### Author Response · Authors · 2026-04-06
> > >
> > > Thank you for the follow-up. We respectfully think the current evidence is already stronger than a merely conceptual justification. For K=1, the paper explicitly states that flat and hierarchical attackers have equal expressive power; therefore, once a consistent gap is observed under matched victim policies, attack budgets, and training budgets, the most natural explanation is optimization difficulty rather than representational advantage. This is exactly the point of Appendix D.
> > >
> > > That said, we agree it would strengthen the paper to diagnose this optimization gap more directly. We will therefore add explicit optimization metrics in the revision, rather than relying only on final attack performance.

---

### Official Review · Reviewer_rDca · 2026-03-16

**Soundness:** 3
**Presentation:** 3
**Significance:** 3
**Originality:** 3
**Overall Recommendation:** 4
**Confidence:** 2

**Summary:**

The authors propose BHEA, a hierarchical adversarial attack for CMARL. BHEA factorises the attack decision into a budget aware victim selector and an action attacker. This decomposition is shown (empirically) to produce stronger attacks under limited episode-level budgets compared to flat policies and other past methods. The authors subsequently introduce an alternating adversarial training framework that expose the victim policy to progressively stronger adversarial behaviours. The victim is then trained against adversaries sampled from this pool, encouraging it to learn coordination-preserving recovery strategies under sparse action hijacking.

**Compliance With Llm Reviewing Policy:**

Affirmed.

**Final Justification:**

I have now updated my score as the additional FLOPs and memory analysis adequately addresses my efficiency concern.
The formal characterisation is not what I hoped for but (so it remains open) the evidence from the K=1 ablation is acceptable .

**Key Questions For Authors:**

1. Can you provide a theoretical or empirical characterisation of when the $\pi_{\mathrm{sel}} \times \pi_{\mathrm{act}}$ factorisation loses
    optimality. For example, settings with tight action-level coupling between simultaneously hijacked agents?

2. Could you report the victim-attacker sample split for ROMANCE and WALL alongside BHEA-AT as well as total FLOPs and peak GPU memory for each method? This would clarify whether Table 2 reflects genuine algorithmic efficiency or differences in resource allocation.

**Limitations:**

yes

**Strengths And Weaknesses:**

Strengths:

The core idea of factorising the adversarial attack into a budget-aware victim selector and an action attacker is well-motivated and cleanly formalised. The equivalence proof between the hierarchical and $(N+1)$-class selector (Section~4.1) seems rigorous to me (but please note low confidence overall). The unseen attack evaluation is a particularly valuable inclusion as it demonstrates that the robustness gains of BHEA-AT are not simply an artefact of overfitting to the training attacker(s).


Weaknesses:

1. Lack of formal justification for the selector-attacker factorization.

The paper proves that the hierarchical $(N+1)$-class victim selector is expressively equivalent to a flat selector but the higher-level factorization $\pi_{\mathrm{adv}} = \pi_{\mathrm{sel}} \times \pi_{\mathrm{act}}$ is assumed without formal analysis.
The two components are optimised with independent value functions and (if I understood correctly) separate GAE advantages. Thus,  implicitly assuming credit assignment decomposes cleanly across decision layers. This may not be the case, in settings where the optimal victim choice depends tightly on the available replacement actions (e.g., environments with strong action-level coupling between simultaneously hijacked agents). In such cases, this decoupling could exclude joint optima reachable by a fully coupled policy. The supporting evidence is purely empirical with no theoretical characterisation provided for when this decomposition is lossy or not.


2. Incomplete computational cost analysis.

The paper claims substantial training efficiency gains but reports only wall-clock hours on a single GPU configuration. This metric conflates several confounding factors: differences in GPU utilisation and FLOPs across methods, the memory footprint (note that BHEA maintains two policy heads and two value networks) and sample allocation across methods (e.g., BHEA-AT dedicates 30\% of total episodes to attacker training, but the analogous split for ROMANCE and WALL is not reported, making it difficult to assess whether the comparison is "like for like"). Moreover, the comparison covers only small SMAC maps but the autoregressive victim selection for $K>1$ could scale differently on larger teams. A more rigorous efficiency comparison would preferably report total FLOPs, peak memory and victim performance as a function of environment steps rather than wall-clock time alone.

---

> ### Author Rebuttal · Authors · 2026-03-26
>
> >### Weakness1&Question1
> Can you provide a theoretical or empirical characterisation of when the $\pi_{sel} \times \pi_{act}$factorisation loses optimality.
>
> ### Answer1.
> We thank the reviewer for this important point. We agree that the paper does not prove that the $\pi_{sel}\times\pi_{act}$ factorization is lossless in all settings: a fully coupled attacker may in principle achieve a better joint optimum, especially when the best victim set depends on the availability of a high-value coordinated replacement action for that set, so victim selection cannot be evaluated independently of downstream action optimization.
> Our goal, however, is not to claim universal optimality, but to improve learnability under the large combinatorial action space. As shown in Appendix D, even for \(K=1\), where flat and hierarchical policies are expressively equivalent, the flat policy is much weaker than BHEA in practice, suggesting that the main bottleneck is optimization rather than expressiveness. For \(K>1\), the flat action space grows combinatorially with \(K\), making fully coupled optimization increasingly impractical. This is exactly why we adopt the hierarchical factorization: it may sacrifice some expressiveness in the worst case, but yields a much more learnable and scalable policy class in practice.
>
> >### Weakness2&Question2
> Could you report the victim-attacker sample split for ROMANCE and WALL alongside BHEA-AT as well as total FLOPs and peak GPU memory for each method?
>
> ### Answer2.
> Thank you for the suggestion. We agree that wall-clock time alone is not sufficient to fully characterize computational efficiency. However, Table 2 was not obtained under unequal training budgets: all adversarial-training methods were run with the same total number of episodes on each SMAC map, and baseline-specific hyperparameters followed the default settings in their original papers. For BHEA-AT, 30% of the total episodes are allocated to attacker training in 10 alternating rounds. For ROMANCE and WALL, their training procedures are not based on the same attacker-victim alternating decomposition, so an exactly analogous split is not directly definable; thus, our fairness criterion is matched total episode budget.
> We agree that FLOPs and peak memory should also be reported. To address this concern, we now provide two additional tables that report, for each method and each SMAC map, the average training FLOPs and peak GPU memory usage measured during training. These results complement the original wall-clock comparison by providing a more complete view of computational cost. As shown in Tables R1–R2, BHEA-AT consistently requires substantially fewer training FLOPs and lower peak GPU memory than WALL across all six SMAC maps, while also remaining much lighter than ROMANCE in memory usage and in most cases in compute cost. This strengthens our original efficiency claim beyond wall-clock time alone.
>
> All methods were evaluated under the same hardware setup and the same total training budget as in the main paper. For each method on each SMAC map, we profiled full training updates within the actual training pipeline and report the average FLOPs per training update. For victim-training methods, this corresponds to one complete learner update; for methods with additional attacker or population-maintenance components (e.g., WALL and BHEA-AT), the profiled update also includes the corresponding extra training logic. FLOPs were measured during training using the PyTorch profiler. Because the profiler reports operator-level estimated FLOPs, these numbers should be interpreted as profiler-estimated training FLOPs rather than exact hardware-level FLOPs. Peak GPU memory was measured using `torch.cuda.max_memory_allocated()`, which records the maximum tensor memory allocated by PyTorch during the profiled training update. The same profiling procedure was applied to all methods to ensure a fair comparison.
> ### Table R1. Average FLOPs per training update on the six SMAC maps
> | Method | 3m | 8m | 2s3z | 3s_vs_3z | 1c3s5z | MMM |
> |---|---:|---:|---:|---:|---:|---:|
> | Vanilla Qmix | 7.401e+08 | 2.114e+09 | 2.135e+09 | 1.687e+09 | 1.276e+10 | 7.112e+09 |
> | Random | 7.401e+08 | 2.114e+09 | 2.135e+09 | 1.687e+09 | 1.276e+10 | 7.112e+09 |
> | ROMANCE | 1.421e+10 | 7.568e+10 | 6.327e+10 | 5.183e+10 | 7.026e+11 | 8.101e+11 |
> | WALL | 1.041e+11 | 1.386e+11 | 1.171e+11  | 1.060e+11 | 1.479e+11 | 1.603e+11 |
> | BHEA-AT | 9.191e+09 | 2.854e+10 | 1.838e+10 | 9.493e+09 | 4.332e+10 | 4.767e+10 |
>
> ### Table R2. Peak GPU memory (GB) on the six SMAC maps
> | Method | 3m | 8m | 2s3z | 3s_vs_3z | 1c3s5z | MMM |
> |---|---:|---:|---:|---:|---:|---:|
> | Vanilla Qmix | 0.033 | 0.076 | 0.051 | 0.032 | 0.089 | 0.092 |
> | Random | 0.033 | 0.076 | 0.051 | 0.032 | 0.089 |  0.092 |
> | ROMANCE | 0.062 | 0.175 | 0.118 | 0.063 | 0.190 | 0.191 |
> | WALL | 0.418 | 0.955  | 0.553 | 0.484 | 1.025 | 1.365 |
> | BHEA-AT | 0.037 | 0.080  | 0.055 | 0.039 | 0.094 | 0.098 |

---

### Official Review · Reviewer_v6oK · 2026-03-17

**Soundness:** 3
**Presentation:** 4
**Significance:** 2
**Originality:** 2
**Overall Recommendation:** 4
**Confidence:** 3

**Summary:**

This paper addresses the vulnerability of cooperative multi-agent reinforcement learning (CMARL) policies to adversarial action hijacking attacks, particularly when attacks are constrained by explicit budgets. The authors claim to analyze a pertinent issue in deploying MARL systems to safety-critical environments, where even sparse adversarial interventions can trigger cascading coordination failures.

The study focuses on the domain of adversarial robustness in CMARL under the CTDE paradigm. The authors observe that existing attack methods—whether planner-based (Wolfpack) or population-based (EGA)—struggle to efficiently exploit limited attack opportunities and incur substantial computational costs.

The main contribution is BHEA (Budgeted Hierarchical Efficient Attack), which decomposes the adversarial policy into: (1) a victim selector deciding when to attack and which agents to hijack, and (2) an action attacker determining replacement actions. This hierarchical structure enables explicit reasoning about attack timing and coordinated victim selection under budget constraints.

Building on BHEA, the authors propose BHEA-AT, an alternating adversarial training framework where victim policies train against adversaries sampled from a pool with recency-biased sampling. Experiments on SMAC and MPE demonstrate that BHEA induces stronger performance degradation than prior methods under matched budgets, while BHEA-AT achieves improved robustness with 35-60% training time reduction compared to ROMANCE and WALL, generalizing across multiple victim algorithms (QMIX, VDN, QPLEX, MAPPO, MADDPG).

**Compliance With Llm Reviewing Policy:**

Affirmed.

**Key Questions For Authors:**

## Key Questions For Authors

**Q1: Threat model realism regarding observation access**

The adversary's observation encoder requires joint observations. How would BHEA perform if the adversary only accesses local observations of compromised agents? Have you experimented with decentralized adversary variants?

*Impact on evaluation: If BHEA remains effective under decentralized adversary constraints, this would strengthen a key soundness concern, potentially raising my significance rating.*


**Q2: Cross-budget generalization**

Figure 2 analyzes attack sensitivity by varying $(K, B)$ against a fixed victim. However, in deployment, defenders cannot anticipate the attacker's exact budget. If BHEA-AT trains with budget $(K_1, B_1)$, how robust is the resulting policy when tested against attackers with different budgets $(K_2, B_2)$? Does robustness transfer across budget regimes?

*Impact on evaluation: Evidence of cross-budget generalization would significantly strengthen the practical applicability claims and could improve my assessment of both soundness and significance.*


**Q3: Scalability to larger agent teams**

All experiments involve 3-10 agents. The hierarchical selector scales linearly in $N$, but coordination complexity often scales super-linearly with team size. Have you evaluated BHEA on scenarios with larger teams (e.g., 20+ agents)? Does the hierarchical advantage over flat policies increase or diminish with scale?

*Impact on evaluation: Evidence of scalability would substantially strengthen significance claims for real-world multi-agent systems, which often involve larger teams than SMAC benchmarks.*

**Limitations:**

No. The paper lacks an explicit limitations section.

# Constructive suggestions

These are all similar points about my question above, would love to see more thinking on limitations from the author.

1. Threat model assumptions: The adversary's access to joint observations $\mathbf{o}_t$ should be explicitly acknowledged as a strong assumption that may not hold in realistic deployment scenarios

2. Scalability: The evaluation is limited to 3-10 agents; the authors should discuss whether the approach scales to larger multi-agent systems

3. Benchmark scope: All experiments use game-based simulations (SMAC, MPE); limitations regarding transfer to real-world domains (multi-robot systems, autonomous vehicles) should be acknowledged

4. Budget specification: The manual tuning of $(K, B)$ per scenario and lack of cross-budget generalization analysis deserves discussion

**Strengths And Weaknesses:**

# Soundness
*Strengths*
1. The hierarchical factorization $\pi_{\text{adv}} = \pi_{\text{sel}} \times \pi_{\text{act}}$ is mathematically well-grounded, and the equivalence to the $(N+1)$-class parameterization is correctly established, enabling tractable PPO optimization while preserving the semantic structure
2. The experimental setup is rigorous: 5 random seeds, matched attack budgets across methods, and evaluation across diverse MARL backbones
3. The ablation study isolating selector vs. attacker contributions and the flat vs. hierarchical comparison provide convincing evidence that performance gains stem from the proposed decomposition rather than confounding factors

*Weaknesses*
1. Testing on unseen attacks addresses a critical concern in adversarial training—that robustness may arise from overfitting to training adversaries rather than genuine coordination resilience
2. The adversary assumes access to joint observations $\mathbf{o}_t$, which contradicts the decentralized execution assumption of CTDE; in realistic threat models, an adversary may only observe compromised agents' local information
3. Budget parameters $(K, B)$ are manually tuned per scenario, yet no sensitivity analysis examines how misspecified budgets during training affect test-time robustness against differently-budgeted attackers. Specifically, whether policies trained with BHEA-AT at budget $(K_1, B_1)$ remain robust against attackers operating with different budgets $(K_2, B_2)$.
4. The claim that hierarchical structure improves optimization is supported empirically but lacks theoretical grounding


# Presentation
*Strengths*
1. The paper maintains clear exposition of a complex multi-level optimization problem; the progression from $K=1$ to $K>1$ cases is pedagogically effective
2. Algorithm boxes and mathematical definitions are precise enough for reproduction

*Weaknesses*
1. The relationship between BHEA-AT and existing population-based methods (ROMANCE) could be more precisely characterized—both maintain adversary pools, but the sampling and update schedules differ in ways that deserve explicit comparison


# Significance
*Strengths*
1. The 35-60% training time reduction over WALL/ROMANCE while achieving superior robustness addresses a genuine bottleneck in adversarial MARL training
2. The observation that hierarchical structure yields better credit assignment than equivalent-capacity flat policies has implications beyond this specific application—it suggests that inductive biases in adversary design matter as much as in agent design

*Weaknesses*
1. Evaluation is restricted to relatively small-scale scenarios (3-10 agents); scalability to larger teams where coordination is more complex remains untested


# Originality
*Strengths*
1. The key insight—that budget-constrained attacks benefit from explicit temporal reasoning about *when* to spend limited resources
2. The sequential victim selection with autoregressive masking elegantly handles the combinatorial $K>1$ case while maintaining linear complexity in $N$

*Weaknesses*
1. While the hierarchical decomposition is novel for this setting, similar when/where/what factorizations appear in hierarchical RL and options frameworks—the connection to this literature is underexplored

---

> ### Author Rebuttal · Authors · 2026-03-29
>
> ### Response to Q1, Soundness W2 and Suggestion 1
> >Thank you for this important comment. We agree that our current attacker uses a stronger observability assumption than the victim’s decentralized execution setting, and we will clarify this more explicitly in the revision. More precisely, CTDE constrains the victim policy to execute from local observations, whereas our attacker is an external runtime adversary defined by the threat model. Therefore, this **does not change the victim’s decentralized execution assumption**, but it does correspond to a stronger attacker access setting than purely local-information attacks.
>
> >We also agree that a more realistic threat model may restrict the attacker to only the local information of compromised agents. To examine this, we additionally evaluate a **restricted-access variant** on 2s3z, 8m, MMM, and 1c3s5z, where the attacker can only observe and attack within a randomly selected subset of 4 agents. This is a strictly weaker setting than full-access BHEA, since both observability and attack scope are reduced. As shown in Table A, the attacker remains effective, although weaker than full access.
>
> **Table A. Victim win rate (%) under full-access BHEA and restricted-access BHEA (subset of 4 agents).**
> |Method|2s3z|8m|1c3s5z|MMM|
> |---|---:|---:|---:|---:|
> |BHEA (full access)|10.2±3.5|11.3±2.5|22.1±5.5|8.1±3.2|
> |BHEA (restricted access)|15.2±3.8|37.2±5.5|42.2±7.7| 26.2±4.1|
> >These results show that stronger team-level observability improves attack strength, but BHEA does not fundamentally rely on it: it still learns non-trivial attacks under substantially more restrictive access. We will clarify this threat-model distinction and include the restricted-access results in the revision.
> ### Response to Q2, Soundness W3 and Suggestion 4
> >To directly address it, we additionally evaluate the BHEA-AT-trained victim under nearby attack configurations around the nominal training setting (B,K): (B+1,K), (B+2,K) , (B,K+1) and (B,K+2).
> This is different from Fig. 2: there we vary B or K to study **attack strength**, while here we test whether the **trained defender remains robust under budget/capacity mismatch**.
> As shown in Table B, the victim remains robust under these nearby perturbations, with performance degrading gracefully as the attacker becomes stronger. This suggests that the learned robustness is not tied to one exact training pair (B,K), but transfers to nearby attack regimes as well.
>
> **Table B. Victim win rate (%) of the same BHEA-AT-trained victim under nearby budget/capacity perturbations around (B,K).**
>
> | Environment |(B+1,K)|(B+2,K)|(B,K)|(B,K+1)|(B,K+2)|
> |---|---:|---:|---:|---:|---:|
> |2s3z|89.1±3.3|73.1±4.6 |95.1±1.6|91.1±5.2|  83.1±3.7|
> |8m|84.7±1.9|70.1±2.8|93.1±2.5|89.1±3.1|79.1±2.6|
> |1c3s5z|91.3±1.6|83.2±1.3|99.2±0.8|94.1±1.4| 90.1±1.9|
> |MMM|79.4±1.6|70.4±2.7|87.3±3.5 |81.1±4.1| 77.1±3.5|
>
> ### Response to Soundness W1
> >This point appears to be a strength rather than a weakness; it may have been placed under the weakness section by mistake.
>
> ### Response to Originality W1 and Presentation W1
> > Thank you for these helpful suggestions. While related ideas appear in hierarchical RL and pool-based training, our contribution is a budget-aware hierarchical attack formulation for cooperative MARL and its corresponding adversarial training scheme. BHEA explicitly decomposes attack timing, victim selection, and action replacement under hard budget constraints, and BHEA-AT uses attacker snapshots as progressively stronger training opponents. We will clarify these distinctions from hierarchical RL/options and ROMANCE in the revision.
>
> ### Response to Q3, Significance W1 and Suggestions 2,3
> > Thank you for these important suggestions. We agree that the current empirical scope should be stated as a limitation. Our experiments cover both SMAC and MPE, but are still limited to relatively small-scale settings (about 3–10 agents), so we do not yet provide direct evidence on larger teams.
> From a method-design perspective, BHEA is intended to scale better than flat attack policies, since it decomposes the attack into timing, victim selection, and action replacement instead of optimizing over a large joint attack space. As discussed in Appendix D, we expect its optimization advantage over flat policies to become more pronounced. Evaluating larger teams and real-world domains is beyond the scope of the current submission, and we will add both points to the limitations discussion in the revision.
>
> ### Response to Soundness W4
> > Thank you for this important comment. We agree that our current support for the optimization benefit of the hierarchical design is empirical rather than theoretical. In the paper, we show that even in the K=1 case, where flat and hierarchical policies are expressively equivalent, the hierarchical version learns stronger attacks in practice. We will clarify this point and avoid overstating the claim.

---

> > ### Author Rebuttal · Reviewer_v6oK · 2026-04-03
> >
> > Thanks for your response and looking forward to the improvement on the paper.

---

### Decision · Program_Chairs · 2026-04-30

**Decision:**

Accept (regular)

**Comment:**

This paper proposes Budgeted Hierarchical Efficient Attack (BHEA), a hierarchical adversarial attack framework for budget-constrained action hijacking in Cooperative multi-agent reinforcement learning (CMARL). Experiments on SMAC demonstrate stronger attacks under matched budgets. The paper makes a technically sound contribution to an important problem, with strong empirical support and a thorough rebuttal. The hierarchical attack formulation and associated adversarial training efficiency gains are likely to be useful to the CMARL robustness community. However, suggested revisions include: (1) explicit limitations section covering threat model assumptions, scalability, and benchmark scope, (2) clarified positioning relative to hierarchical RL and self-play literature, (3) the restricted-access and cross-budget experiments from the rebuttal, and (4) computational cost comparisons (FLOPs, memory) alongside wall-clock time.